# A solution to the learning dilemma for recurrent networks of spiking neurons

Guillaume Bellec[1,2], Franz Scherr [1,2], Anand Subramoney [1], Elias Hajek[1], Darjan Salaj [1], Robert Legenstein [1] & Wolfgang Maass [1✉]

Recurrently connected networks of spiking neurons underlie the astounding information processing capabilities of the brain. Yet in spite of extensive research, how they can learn through synaptic plasticity to carry out complex network computations remains unclear. We argue that two pieces of this puzzle were provided by experimental data from neuroscience. A mathematical result tells us how these pieces need to be combined to enable biologically plausible online network learning through gradient descent, in particular deep reinforcement learning. This learning method–called e-prop–approaches the performance of back-propagation through time (BPTT), the best-known method for training recurrent neural networks in machine learning. In addition, it suggests a method for powerful on-chip learning in energy-efficient spike-based hardware for artificial intelligence.

[1] Institute of Theoretical Computer Science, Graz University of Technology, Inffeldgasse 16b, Graz, Austria. [2] These authors contributed equally: Guillaume Bellec, Franz Scherr. ✉email: maass@igi.tugraz.at

Networks of neurons in the brain differ in at least two essential aspects from deep neural networks in machine learning: they are recurrently connected, forming a giant number of loops, and they communicate via asynchronously emitted stereotypical electrical pulses, called spikes, rather than bits or numbers that are produced in a synchronized manner by each layer of a deep feedforward network. We consider the arguably most prominent model for spiking neurons in the brain: leaky integrate-and-fire (LIF) neurons, where spikes that arrive from other neurons through synaptic connections are multiplied with the corresponding synaptic weight, and are linearly integrated by a leaky membrane potential. The neuron fires—i.e., emits a spike—when the membrane potential reaches a firing threshold.

But it is an open problem how recurrent networks of spiking neurons (RSNNs) can learn, i.e., how their synaptic weights can be modified by local rules for synaptic plasticity so that the computational performance of the network improves. In deep learning, this problem is solved for feedforward networks through gradient descent for a loss function $E$ that measures imperfections of current network performance[1]. Gradients of $E$ are propagated backwards through all layers of the feedforward network to each synapse through a process called backpropagation. Recurrently connected networks can compute more efficiently because each neuron can participate several times in a network computation, and they are able to solve tasks that require integration of information over time or a non-trivial timing of network outputs according to task demands. Therefore, the impact of a synaptic weight on the loss function (see Fig. 1a) is more indirect, and learning through gradient descent becomes substantially more difficult. This problem is aggravated if there are slowly changing hidden variables in the neuron model, as in neurons with spike-frequency adaptation (SFA). Neurons with SFA are quite common in the neocortex[2], and it turns out that their inclusion in the RSNN significantly increases the computational power of the network[3]. In fact, RSNNs trained through gradient descent acquire then similar computing capabilities as networks of LSTM (long short-term memory) units, the state of the art for recurrent neural networks in machine learning. Because of this functional relation to LSTM networks these RSNN models are referred to as LSNNs[3].

In machine learning, one trains recurrent neural networks by unrolling the network into a virtual feedforward network[1], see Fig. 1b, and applying the backpropagation algorithm to that (Fig. 1c). This method is called backpropagation through time (BPTT), as it requires propagation of gradients backwards in time.

With a careful choice of the pseudo derivative for handling the discontinuous dynamics of spiking neurons, one can apply BPTT also to RSNNs, and RSNNs were able to learn in this way to solve really demanding computational tasks[3,4]. But the dilemma is that BPTT requires storing the intermediate states of all neurons during a network computation, and merging these in a subsequent offline process with gradients that are computed backwards in time (see Fig. 1c). This makes it very unlikely that BPTT is used by the brain[5].

The previous lack of powerful online learning methods for RSNNs also affected the use of neuromorphic computing hardware, which aims at a drastic reduction in the energy consumption of AI implementations. A substantial fraction of this neuromorphic hardware, such as SpiNNaker[6] or Intel's Loihi chip[7], implements RSNNs. Although it does not matter here whether the learning algorithm is biologically plausible, the excessive storage and offline processing demands of BPTT make this option unappealing. Hence, there also exists a learning dilemma for RSNNs in neuromorphic hardware.

We are not aware of previous work on online gradient descent learning methods for RSNNs, neither for supervised learning nor for reinforcement learning (RL). There exists, however, preceding work on online approximations of gradient descent for non-spiking neural networks based on[8], which we review in the Discussion Section.

Two streams of experimental data from neuroscience provide clues about the organization of online network learning in the brain:

First, neurons in the brain maintain traces of preceding activity on the molecular level, for example, in the form of calcium ions or activated CaMKII enzymes[9]. In particular, they maintain a fading memory of events where the presynaptic neuron fired before the postsynaptic neuron, which is known to induce synaptic plasticity if followed by a top–down learning signal[10–12]. Such traces are often referred to as eligibility traces.

Second, in the brain, there exists an abundance of top–down signals such as dopamine, acetylcholine, and neural firing[13] related to the error-related negativity, that inform local populations of neurons about behavioral results. Furthermore, dopamine signals[14,15] have been found to be specific for different target populations of neurons, rather than being global. We refer in our learning model to such top–down signals as learning signals.

A re-analysis of the mathematical basis of gradient descent learning in recurrent neural networks tells us how local eligibility traces and top–down learning signals should be optimally combined—without requiring backprogation of signals through time. The resulting learning method e-prop is illustrated in Fig. 1d. It learns slower than BPTT, but tends to approximate the performance of BPTT, thereby providing a first solution to the learning dilemma for RSNNs. Furthermore, e-prop also works for RSNNs with more complex neuron models, such as LSNNs. This new learning paradigm elucidates how the brain could learn to recognize phonemes in spoken language, solve temporal credit assignment problems, and acquire new behaviors just from rewards.

## Results

**Mathematical basis for e-prop.** Spikes are modeled as binary variables $z_j^t$ that assume value 1 if neuron $j$ fires at time $t$, otherwise value 0. It is common in models to let $t$ vary over small discrete time steps, e.g., of 1 ms length. The goal of network learning is to find synaptic weights $W$ that minimize a given loss function $E$. $E$ may depend on all or a subset of the spikes in the network. $E$ measures in the case of regression or classification learning the deviation of the actual output $y_k^t$ of each output neuron $k$ at time $t$ from its given target value $y_k^{*,t}$ (Fig. 1a). In RL, the goal is to optimize the behavior of an agent in order to maximize obtained rewards. In this case, $E$ measures deficiencies of the current agent policy to collect rewards.

The gradient $\frac{dE}{dW_{ji}}$ for the weight $W_{ji}$ of the synapse from neuron $i$ to neuron $j$ tells us how this weight should be changed in order to reduce $E$. It can in principle be estimated—in spite of the fact that the implicit discrete variable $z_j^t$ is non-differentiable—with the help of a suitable pseudo derivative for spikes as in refs. [3,4]. The key innovation is a rigorous proof (see "Methods") that the gradient $\frac{dE}{dW_{ji}}$ can be represented as a sum of products over the time steps $t$ of the RSNN computation, where the second factor is just a local gradient that does not depend on $E$:

$$\frac{dE}{dW_{ji}} = \sum_t \frac{dE}{dz_j^t} \cdot \left[\frac{dz_j^t}{dW_{ji}}\right]_{\text{local}}. \qquad (1)$$

This local gradient is defined as a sum of products of partial derivatives concerning the hidden state $\mathbf{h}_j^t$ of neuron $j$ at time $t$

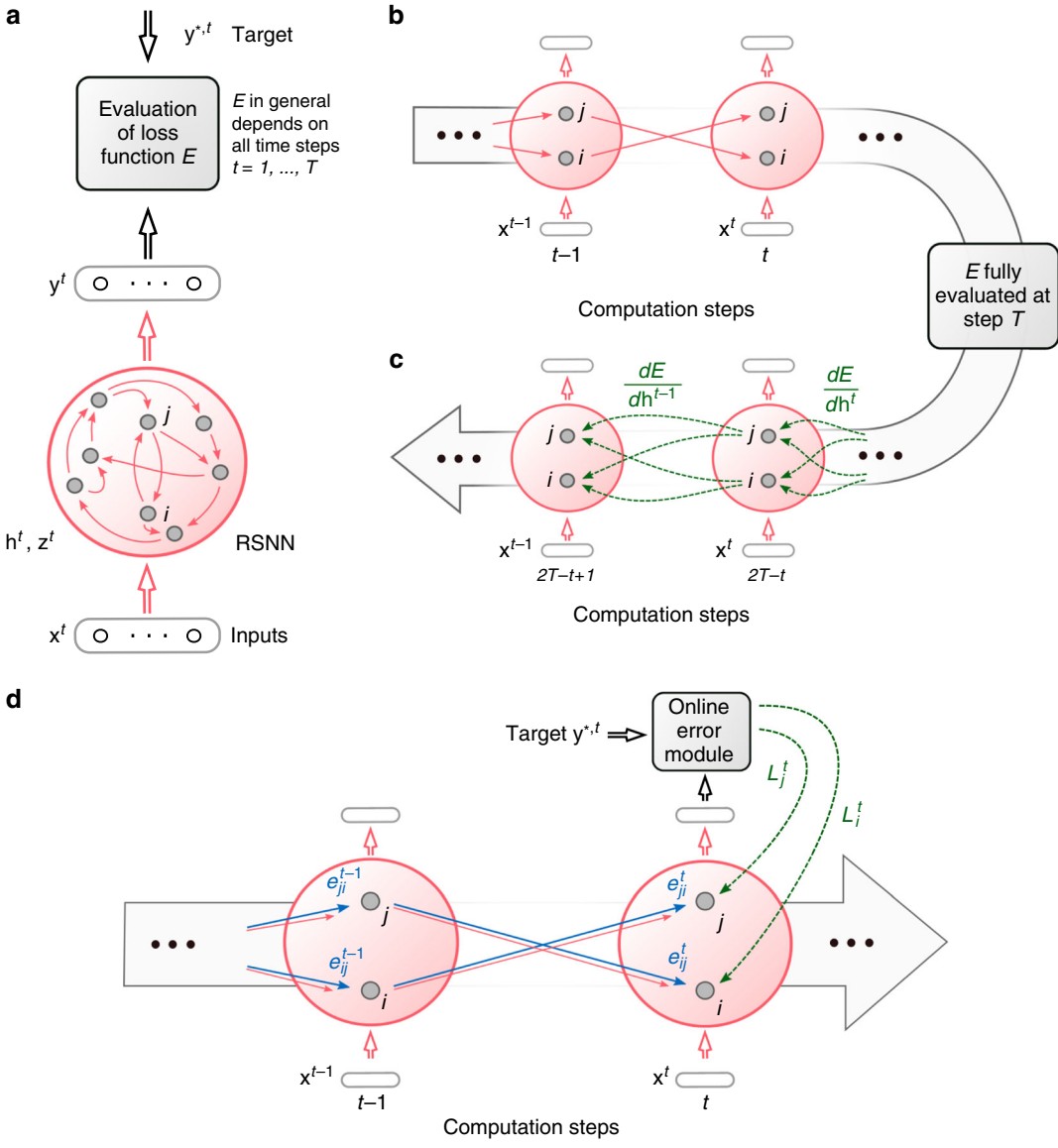

**Fig. 1 Schemes for BPTT and e-prop. a** RSNN with network inputs **x**, neuron spikes **z**, hidden neuron states **h**, and output targets **y***, for each time step $t$ of the RSNN computation. Output neurons **y** provide a low-pass filter of a weighted sum of network spikes **z**. **b** BPTT computes gradients in the unrolled version of the network. It has a new copy of the neurons of the RSNN for each time step $t$. A synaptic connection from neuron $i$ to neuron $j$ of the RSNN is replaced by an array of feedforward connections, one for each time step $t$, which goes from the copy of neuron $i$ in the layer for time step $t$ to a copy of neuron $j$ in the layer for time step $t + 1$. All synapses in this array have the same weight: the weight of this synaptic connection in the RSNN. **c** Loss gradients of BPTT are propagated backwards in time and retrograde across synapses in an offline manner, long after the forward computation has passed a layer. **d** Online learning dynamics of e-prop. Feedforward computation of eligibility traces is indicated in blue. These are combined with online learning signals according to Eq. (1).

and preceding time steps, which can be updated during the forward computation of the RNN by a simple recursion (Eq. (14)). This term $\left[\frac{dz_j^t}{dW_{ji}}\right]_{\text{local}}$ is not an approximation. Rather, it collects the maximal amount of information about the network gradient $\frac{dE}{dW_{ji}}$ that can be computed locally in a forward manner. Therefore, it is the key-factor of e-prop. As it reduces for simple neuron models—whose internal state is fully captured by its membrane potential—to a variation of terms that are commonly referred to as eligibility traces for synaptic plasticity[12], we also refer to

as eligibility trace. But most biological neurons have additional hidden variables that change on a slower time scale, such as the firing threshold of a neuron with firing threshold adaptation. Furthermore, these slower processes in neurons are essential for attaining with spiking neurons similarly powerful computing capabilities as LSTM networks[3]. Hence, the form that this eligibility trace $e_{ji}^t$ takes for adapting neurons (see Eq. (25)) is essential for understanding e-prop, and it is the main driver behind the resulting qualitative jump in computing capabilities of RSNNs, which are attainable through biologically plausible learning. Eqs. (1) and (2) yield the representation

$$e_{ji}^t \stackrel{\text{def}}{=} \left[\frac{dz_j^t}{dW_{ji}}\right]_{\text{local}} \quad (2)$$

$$\frac{dE}{dW_{ji}} = \sum_t L_j^t e_{ji}^t \quad (3)$$

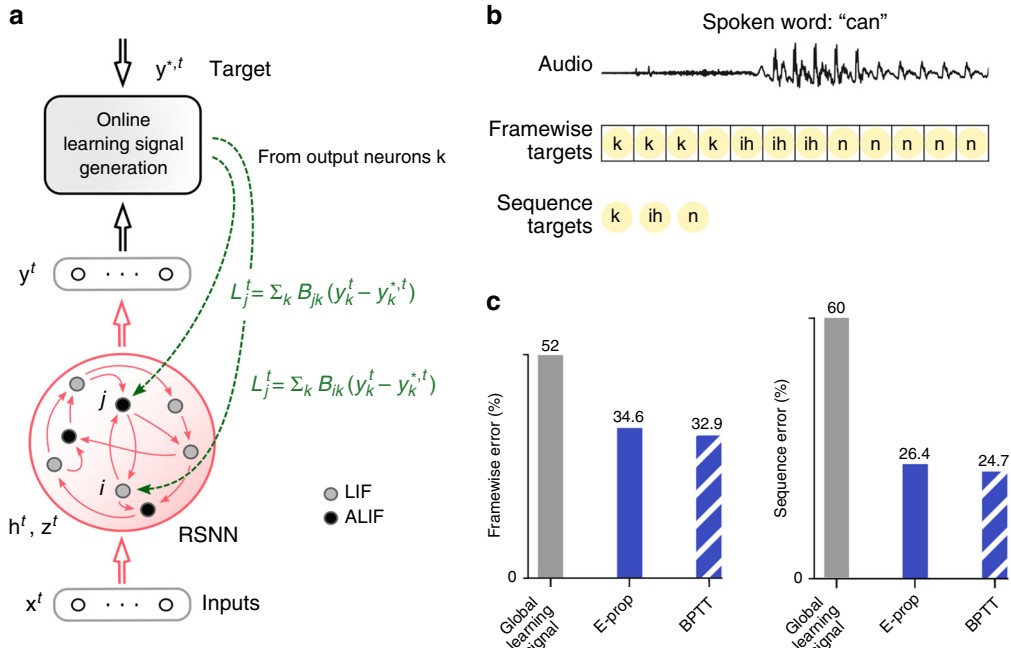

**Fig. 2 Comparison of BPTT and e-prop for learning phoneme recognition. a** Network architecture for e-prop, illustrated for an LSNN consisting of LIF and ALIF neurons. **b** Input and target output for the two versions of TIMIT. **c** Performance of BPTT and symmetric e-prop for LSNNs consisting of 800 neurons for framewise targets and 2400 for sequence targets (random and adaptive e-prop produced similar results, see Supplementary Fig. 2). To obtain the Global learning signal baselines, the neuron-specific feedbacks are replaced with global ones.

of the loss gradient, where we refer to $L_j^t \overset{\text{def}}{=} \frac{dE}{dz_j^t}$ as the learning signal for neuron $j$. This equation defines a clear program for approximating the network loss gradient through local rules for synaptic plasticity: change each weight $W_{ji}$ at step $t$ proportionally to $-L_j^t e_{ji}^t$, or accumulate these so-called tags in a hidden variable that is translated occasionally into an actual weight change. Hence, e-prop is an online learning method in a strict sense (see Fig. 1d). In particular, there is no need to unroll the network as for BPTT.

As the ideal value $\frac{dE}{dz_j^t}$ of the learning signal $L_j^t$ also captures influences that the current spike output $z_j^t$ of neuron $j$ may have on $E$ via future spikes of other neurons, its precise value is in general not available at time $t$. We replace it by an approximation, such as $\frac{\partial E}{\partial z_j^t}$, which ignores these indirect influences (this partial derivative $\frac{\partial E}{\partial z_j^t}$ is written with a rounded $\partial$ to signal that it captures only the direct influence of the spike $z_j^t$ on the loss function $E$). This approximation takes only currently arising losses at the output neurons $k$ of the RSNN into account, and routes them with neuron-specific weights $B_{jk}$ to the network neurons $j$ (see Fig. 2a):

$$L_j^t = \sum_k B_{jk} \underbrace{(y_k^t - y_k^{*,t})}_{\substack{\text{deviation of output } k \\ \text{at time } t}}. \tag{4}$$

Although this approximate learning signal $L_j^t$ only captures errors that arise at the current time step $t$, it is combined in Eq. (3) with an eligibility trace $e_{ji}^t$ that may reach far back into the past of neuron $j$ (see Fig. 3b), thereby alleviating the need to solve the temporal credit assignment problem by propagating signals backwards in time (like in BPTT).

There are several strategies for choosing the weights $B_{jk}$ for this online learning signal. In symmetric e-prop, we set it equal to the corresponding weight $W_{kj}^{\text{out}}$ of the synaptic connection from

neuron $j$ to output neuron $k$, as demanded by $\frac{\partial E}{\partial z_j^t}$. Note that this learning signal would actually implement $\frac{dE}{dz_j^t}$ exactly in the absence of recurrent connections in the network. Biologically more plausible are two variants of e-prop that avoid weight sharing: in random e-prop, the values of all weights $B_{jk}$—even for neurons $j$ that are not synaptically connected to output neuron $k$—are randomly chosen and remain fixed, similar to Broadcast Alignment for feedforward networks[16–18]. In adaptive e-prop, in addition to using random backward weights, we also let $B_{jk}$ evolve through a simple local plasticity rule that mirrors the plasticity rule applied to $W_{kj}^{\text{out}}$ for neurons $j$ that are synaptically connected to output neuron $k$ (see Supplementary Note 2).

Resulting synaptic plasticity rules (see Methods) look similar to previously proposed plasticity rules[12] for the special case of LIF neurons without slowly changing hidden variables. In particular, they involve postsynaptic depolarization as one of the factors, similarly as the data-based Clopath-rule in ref. [19], see Supplementary Note 6 for an analysis.

**Learning phoneme recognition with e-prop**. The phoneme recognition task TIMIT[20] is one of the most commonly used benchmarks for temporal processing capabilities of different types of recurrent neural networks and different learning approaches[21]. It comes in two versions. Both use, as input, acoustic speech signals from sentences that are spoken by 630 speakers from eight dialect regions of the USA (see the top of Fig. 2b for a sample segment). In the simpler version, used for example in ref. [21], the goal is to recognize which of 61 phonemes is spoken in each 10 ms time frame (framewise classification). In the more-sophisticated version from ref. [22], which achieved an essential step toward human-level performance in speech-to-text transcription, the goal is to recognize the sequence of phonemes in the entire spoken sentence independently of their timing (sequence transcription). RSNNs consisting only of LIF neurons do not even reach good performance on TIMIT with BPTT[3].

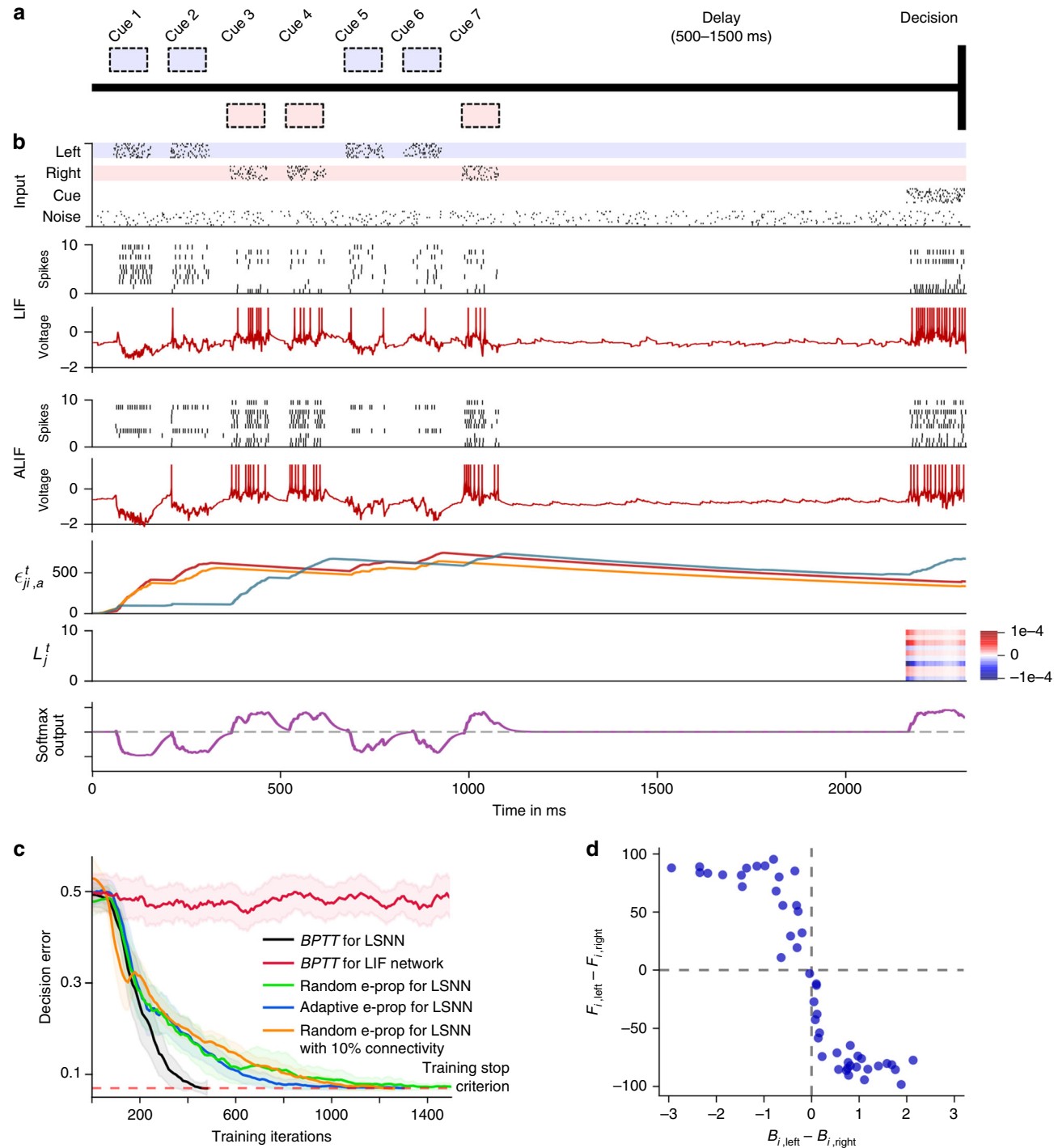

**Fig. 3 Solving a task with difficult temporal credit assignment. a** Setup of corresponding rodent experiments of ref. [23] and ref. [14], see Supplmentary Movie 1. **b** Input spikes, spiking activity of 10 out of 50 sample LIF neurons and 10 out of 50 sample ALIF neurons, membrane potentials (more precisely: $v_j^t - A_j^t$) for two sample neurons $j$, three samples of slow components of eligibility traces, sample learning signals for 10 neurons and softmax network output. **c** Learning curves for BPTT and two e-prop versions applied to LSNNs, and BPTT applied to an RSNN without adapting neurons (red curve). Orange curve shows learning performance of e-prop for a sparsely connected LSNN, consisting of excitatory and inhibitory neurons (Dale's law obeyed). The shaded areas are the 95% confidence intervals of the mean accuracy computed with 20 runs. **d** Correlation between the randomly drawn broadcast weights $B_{jk}$ for $k =$ left/right for learning signals in random e-prop and resulting sensitivity to left and right input components after learning. $f_{j,\text{left}}$ ($f_{j,\text{right}}$) was the resulting average firing rate of neuron $j$ during presentation of left (right) cues after learning.

Hence, we are considering here LSNNs, where a random subset of the neurons is a variation of the LIF model with firing rate adaptation (adaptive LIF (ALIF) neurons), see Methods. The name LSNN is motivated by the fact that this special case of the RSNN model can achieve through training with BPTT similar performance as an LSTM network[3].

E-prop approximates the performance of BPTT on LSNNs for both versions of TIMIT very well, as shown in Fig. 2c.

Furthermore, LSNNs could solve the framewise classification task without any neuron firing more frequently than 12 Hz (spike count taken over 32 spoken sentences), demonstrating that they operate in an energy-efficient spike-coding—rather than a rate coding—regime. For the more difficult version of TIMIT, we trained as in ref. [22], a complex LSNN consisting of a feedforward sequence of three recurrent networks. Our results show that e-prop can also handle learning for such more complex network structures very well. In Supplementary Fig. 4 we show for comparison also the performance of e-prop and BPTT for LSTM networks on the same tasks. These data show that for both versions of TIMIT the performance of e-prop for LSNNs comes rather close to that of BPTT for LSTM networks. In addition, they show that e-prop also provides for LSTM networks a functionally powerful online learning method.

**Solving difficult temporal credit assignment**. A hallmark of cognitive computations in the brain is the capability to go beyond a purely reactive mode: to integrate diverse sensory cues over time, and to wait until the right moment arrives for an action. A large number of experiments in neuroscience analyze neural coding after learning such tasks (see e.g., refs. [14,23]). But it had remained unknown how one can model the learning of such cognitive computations in RSNNs of the brain. In order to test whether e-prop can solve this problem, we considered the same task that was studied in the experiments of ref. [23] and ref. [14]. There a rodent moved along a linear track in a virtual environment, where it encountered several visual cues on the left and right, see Fig. 3a and Supplementary Movie 1. Later, when it arrived at a T-junction, it had to decide whether to turn left or right. It was rewarded when it turned to that side from which it had previously received the majority of visual cues. This task is not easy to learn as the subject needs to find out that it does not matter on which side the last cue was, or in which order the cues were presented. Instead, the subject has to learn to count cues separately for each side and to compare the two resulting numbers. Furthermore, the cues need to be processed properly long before a reward is given. We show in Supplementary Fig. 5 that LSNNs can learn this task via e-prop in exactly the same way just from rewards. But as the way how e-prop solves the underlying temporal credit assignment problem is easier to explain for the supervised learning version of this task, we discuss here the case where a teacher tells the subject at the end of each trial what would have been the right decision. This still yields a challenging scenario for any online learning method since non-zero learning signals $L_j^t$ arise only during the last 150 ms of a trial (Fig. 3b). Hence, all synaptic plasticity has to take place during these last 150 ms, long after the input cues have been processed. Nevertheless, e-prop is able to solve this learning problem, see Fig. 3c and Supplementary Movie 2. It just needs a bit more time to reach the same performance level as offline learning via BPTT (see Supplementary Movie 3). Whereas this task cannot even be solved by BPTT with a regular RSNN that has no adapting neurons (red curve in Fig. 3c), all three previously discussed variations of e-prop can solve it if the RSNN contains adapting neurons. We explain in Supplementary Note 2 how this task can also be solved by sparsely connected LSNNs consisting of excitatory and inhibitory neurons: by integrating stochastic rewiring[24] into e-prop.

But how can the neurons in the LSNN learn to record and count the input cues if all the learning signals are identically 0 until the last 150 ms of a 2250 ms long trial (see 2nd to last row of Fig. 3b)?

For answering this question, one should note that firing of a neuron $j$ at time $t$ can affect the loss function $E$ at a later time

point $t' > t$ in two different ways: via route (i) it affects future values of slow hidden variables of neuron $j$ (e.g., its firing threshold), which may then affect the firing of neuron $j$ at $t'$, which in turn may directly affect the loss function at time $t'$. Via route (ii) it affects the firing of other neurons $j'$ at $t'$, which directly affects the loss function at time $t'$.

In symmetric and adaptive e-prop, one uses the partial derivative $\frac{\partial E}{\partial z_j^t}$ as learning signal $L_j^t$ for e-prop—instead of the total derivative $\frac{dE}{dz_j^t}$, which is not available online. This blocks the flow of gradient information along route (ii). But the eligibility trace keeps the flow along route (i) open. Therefore, even symmetric and adaptive e-prop can solve the temporal credit assignment problem of Fig. 3 through online learning: the gradient information that flows along route (i) enables neurons to learn how to process the sensory cues at time points t during the first 1050 ms, although this can affect the loss only at time points $t' > 2100$ ms when the loss becomes non-zero.

This is illustrated in the 3rd last row of Fig. 3b: the slow component $\epsilon_{ji,a}^t$ of the eligibility traces $e_{ji}$ of adapting neurons $j$ decays with the typical long time constant of firing rate adaptation (see Eq. (24) and Supplementary Movie 2). As these traces stretch from the beginning of the trial into its last phase, they enable learning of differential responses to left and right input cues that arrived over 1050 ms before any learning signals become non-zero, as shown in the 2nd to last row of Fig. 3b. Hence, eligibility traces provide so-called highways into the future for the propagation of gradient information. These can be seen as biologically realistic replacements for the highways into the past that BPTT employs during its backwards pass (see Supplementary Movie 3).

This analysis also tells us when symmetric e-prop is likely to fail to approximate the performance of BPTT: if the forward propagation of gradients along route (i) cannot reach those later time points $t'$ at which the value of the loss function becomes salient. One can artificially induce this in the experiment of Fig. 3 by adding to the LSNN—which has the standard architecture shown in Fig. 2a—hidden layers of a feedforward SNN through which the communication between the LSNN and the readout neurons has to flow. The neurons $j'$ of these hidden layers block route (i), whereas leaving route (ii) open. Hence, the task of Fig. 3 can still be learnt with this modified network architecture by BPTT, but not by symmetric e-prop, see Supplementary Fig. 8.

Identifying tasks where the performance of random e-prop stays far behind that of BPTT is more difficult, as error signals are sent there also to neurons that have no direct connections to readout neurons. For deep feedforward networks it has been shown in ref. [25] that Broadcast Alignment, as defined in refs. [17,18], cannot reach the performance of Backprop for difficult image classification tasks. Hence, we expect that random e-prop will exhibit similar deficiencies with deep feedforward SNNs on difficult classification tasks. We are not aware of corresponding demonstrations of failures of Broadcast Alignment for artificial RNNs, although they are likely to exist. Once they are found, they will probably point to tasks where random e-prop fails for RSNNs. Currently, we are not aware of any.

Figure 3d provides insight into the functional role of the randomly drawn broadcast weights in random e-prop: the difference of these weights determines for each neuron $j$ whether it learns to respond in the first phase of a trial more to cues from the left or right. This observation suggests that neuron-specific learning signals for RSNNs have the advantage that they can create a diversity of feature detectors for task-relevant network inputs. Hence, a suitable weighted sum of these feature detectors is later able to cancel remaining errors at the network output, similarly as in the case of feedforward networks[16].

We would like to point out that the use of the familiar actor-critic method in reward-based e-prop, which we will discuss in the next section, provides an additional channel by which information about future losses can gate synaptic plasticity of the e-prop learner at the current time step $t$: through the estimate $V(t)$ of the value of the current state, that is simultaneously learnt via internally generated reward prediction errors.

**Reward-based e-prop**. Deep RL has significantly advanced the state of the art in machine learning and AI through clever applications of BPTT to RL[26]. We found that one of the arguably most powerful RL methods within the range of deep RL approaches that are not directly biologically implausible, policy gradient in combination with actor-critic, can be implemented with e-prop. This yields the biologically plausible and hardware friendly deep RL algorithm reward-based e-prop. The LSNN learns here both an approximation to the value function (the critic) and a stochastic policy (the actor). Neuron-specific learning signals are combined in reward-based e-prop with a global signal that transmits reward prediction errors (Fig. 4b). In contrast to the supervised case, where the learning signals $L_j^t$ depend on the deviation from an external target signal, the learning signals communicate here how a stochastically chosen action deviates from the action mean that is currently proposed by the network.

In such RL tasks, the learner needs to explore its environment, and find out which action gets rewarded in what state[27]. There is no teacher that tells the learner what action would be optimal; in fact, the learner may never find that out. Nevertheless, learning methods such as BPTT are essential for a powerful form of RL that is often referred to as Deep RL[26]. There, one trains recurrent artificial neural networks with internally generated teaching signals. We show here that Deep RL can in principle also be carried out by neural networks of the brain, as e-prop approximates the performance of BPTT also in this RL context. However, another new ingredient is needed to prove that. Previous work on Deep RL for solving complex tasks, such as winning Atari games[26], required additional mechanisms to avoid well-known instabilities that arise from using nonlinear function approximators, such as the use of several interacting learners in parallel. As this parallel learning scheme does not appear to be biologically plausible, we introduce here a new method for avoiding learning instabilities: we show that a suitable schedule for the lengths of learning episodes and learning rates also alleviates learning instabilities in Deep RL.

We propose an online synaptic plasticity rule (5) for deep RL, which is similar to equation (3), except that a fading memory filter $\mathcal{F}_\gamma$ is applied here to the term $L_j^t \bar{e}_{ji}^t$, where $\gamma$ is the given discount factor for future rewards and $\bar{e}_{ji}^t$ denotes a low-pass filtered copy of the eligibility trace $e_{ji}^t$ (see Methods). This term is multiplied in the synaptic plasticity rule with the reward prediction error $\delta^t = r^t + \gamma V^{t+1} - V^t$, where $r^t$ is the reward received at time $t$. This yields an instantaneous weight change of the form:

$$\Delta W_{ji}^t = -\eta\, \delta^t \mathcal{F}_\gamma\left(L_j^t \bar{e}_{ji}^t\right). \tag{5}$$

Previous three-factor learning rules for RL were usually of the form $\Delta W^t = \eta \delta^t \bar{e}_{ji}^t$[12,28]. Hence, they estimated gradients of the policy just by correlating the output of network neurons with the reward prediction error. The learning power of this approach is known to be quite limited owing to high noise in the resulting gradient estimates. In contrast, in the plasticity rule (5) for reward-based e-prop, the eligibility traces are first combined with a neuron-specific feedback $L_j^t$, before they are multiplied with the reward prediction error $\delta^t$. We show in Methods analytically that this yields estimates of policy- and value gradients similarly as in deep RL with BPTT. Furthermore, in contrast to previously proposed three-factor learning rules, this rule (5) is also applicable to LSNNs.

We tested reward-based e-prop on a classical benchmark task[26] for learning intelligent behavior from rewards: winning Atari video games provided by the Arcade Learning Environment[29]. To win such game, the agent needs to learn to extract salient information from the pixels of the game screen, and to infer the value of specific actions, even if rewards are obtained in a distant future. In fact, learning to win Atari games is a serious challenge for RL even in machine learning[26]. Besides, artificial neural networks and BPTT, previous solutions also required experience replay (with a perfect memory of many frames and action sequences that occurred much earlier) or an asynchronous training of numerous parallel agents sharing synaptic weight updates. We show here that also an LSNN can learn via e-prop to win Atari games, through online learning of a single agent. This becomes possible with a single agent and without episode replay if the agent uses a schedule of increasing episode lengths—with a learning rate that is inversely related to that length. Using this scheme, an agent can experience diverse and uncorrelated short episodes in the first phase of learning, producing useful skills. Subsequently, the agent can fine-tune its policy using longer episodes.

First, we considered the well-known Atari game Pong (Fig. 4a). Here, the agent has to learn to hit a ball in a clever way using up and down movements of his paddle. A reward is obtained if the opponent cannot catch the ball. We trained an agent using reward-based e-prop for this task, and show a sample trial in Fig. 4c and Supplementary Movie 4. In contrast to common deep RL solutions, the agent learns here in a strict online manner, receiving at any time just the current frame of the game screen. In Fig. 4d, we demonstrate that also this biologically realistic learning approach leads to a competitive score.

If one does not insist on an online setting where the agent receives just the current frame of the video screen but the last four frames, winning strategies for about half of the Atari games can already be learnt by feedforward neural networks (see Supplementary Table 3 of ref. [26]). However, for other Atari games, such as Fishing Derby (Fig. 5a), it was even shown in ref. [26] that deep RL applied to LSTM networks achieves a substantially higher score than any deep RL method for feedforward networks, which was considered there. Hence, in order to test the power of online reward-based e-prop also for those Atari games that require enhanced temporal processing, we tested it on the Fishing Derby game. In this game, the agent has to catch as many fish as possible while avoiding that the shark touches the fish with any part of its body, and that the opponent catches the fish first. We show in Fig. 5c that online reward-based e-prop applied to an LSNN does in fact reach the same performance as reference offline algorithms applied to LSTM networks. We show a random trial after learning in Fig. 5d, where we can identify two different learnt behaviors: first, by evading the shark, and a second by collecting fish. The agent has learnt to switch between these two behaviors as required by the situation.

In general, we conjecture that variants of reward-based e-prop will be able to solve most deep RL tasks that can be solved by online actor-critic methods in machine learning.

## Discussion

We propose that in order to understand the computational function and neural coding of neural networks in the brain, one needs to understand the organization of the plasticity

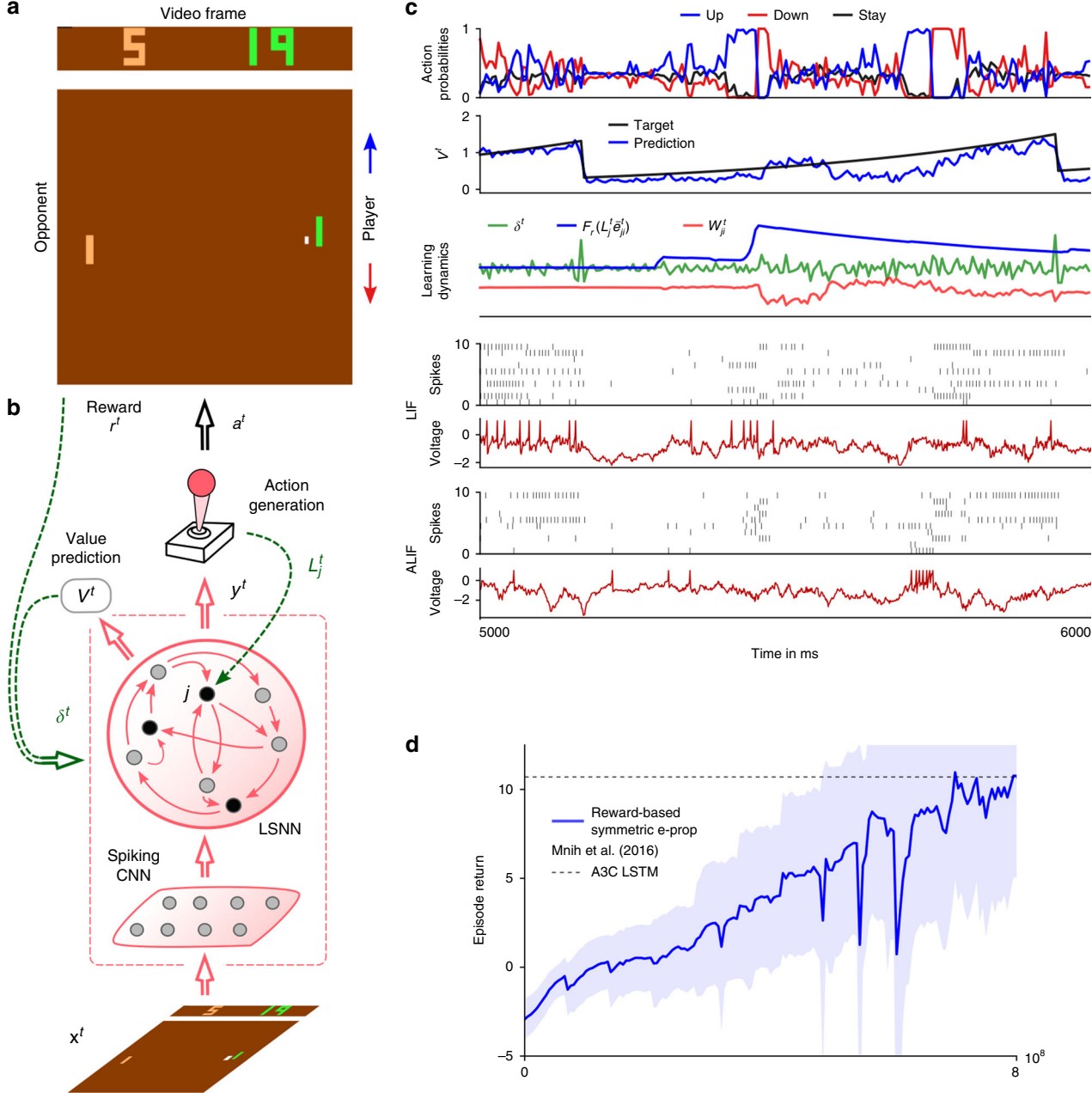

**Fig. 4 Application of e-prop to the Atari game Pong. a** Here, the player (green paddle) has to outplay the opponent (light brown). A reward is acquired when the opponent cannot bounce back the ball (a small white square). To achieve this, the agent has to learn to hit the ball also with the edges of his paddle, which causes a less predictable trajectory. **b** The agent is realized by an LSNN. The pixels of the current video frame of the game are provided as input. During processing of the stream of video frames by the LSNN, actions are generated by the stochastic policy in an online manner. At the same time, future rewards are predicted. The current error in prediction is fed back both to the LSNN and the spiking CNN that preprocesses the frames. **c** Sample trial of the LSNN after learning with reward-based e-prop. From top to bottom: probabilities of stochastic actions, prediction of future rewards, learning dynamics of a random synapse (arbitrary units), spiking activity of 10 out of 240 sample LIF neurons and 10 out of 160 sample ALIF neurons, and membrane potentials (more precisely: $v_j^t - A_j^t$) for the two sample neurons $j$ at the bottom of the spike raster above. **d** Learning progress of the LSNN trained with reward-based e-prop, reported as the sum of collected rewards during an episode. The learning curve is averaged over five different runs and the shaded area represents the standard deviation. More information about the comparison between our results and A3C are given in Supplementary Note 5.

mechanisms that install and maintain these. So far, BPTT was the only candidate for that, as no other learning method provided sufficiently powerful computational function to RSNN models. But as BPTT is not viewed to be biologically realistic[5], it does not help us to understand learning in the brain. E-prop offers a solution to this dilemma, as it does not require biologically unrealistic mechanisms, but still enables RSNNs to learn difficult

computational tasks, in fact almost as well as BPTT. Furthermore, it enables RSNNs to solve these tasks in an energy-efficient sparse firing regime, rather than resorting to rate coding.

E-prop relies on two types of signals that are abundantly available in the brain, but whose precise role for learning have not yet been understood: eligibility traces and learning signals. As e-prop is based on a transparent mathematical principle (see Eq.

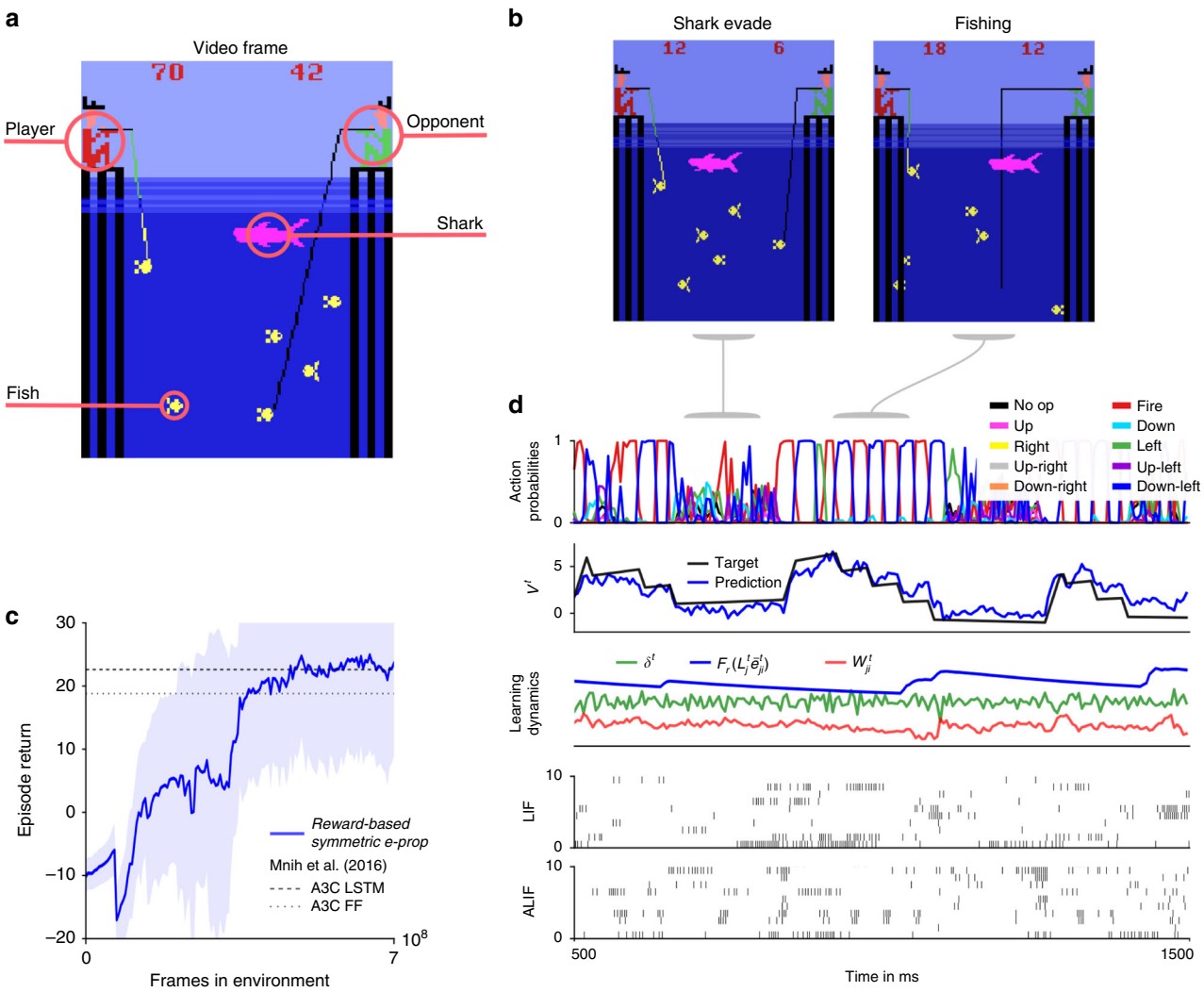

**Fig. 5 Application of e-prop to learning to win the Atari game Fishing Derby. a** Here the player has to compete against an opponent, and try to catch more fish from the sea. **b** Once a fish has bit, the agent has to avoid that the fish gets touched by a shark. **c** Sample trial of the trained network. From top to bottom: probabilities of stochastic actions, prediction of future rewards, learning dynamics of a random synapse (arbitrary units), spiking activity of 20 out of 180 sample LIF neurons and 20 out of 120 sample ALIF neurons. **d** Learning curves of an LSNN trained with reward-based e-prop as in Fig. 4d.

(3)), it provides a normative model for both types of signals, as well as for synaptic plasticity rules. Interestingly, the resulting learning model suggests that a characteristic aspect of many biological neurons—the presence of slowly changing hidden variables—provides a possible solution to the problem how a RSNN can learn without error signals that propagate backwards in time: slowly changing hidden variables of neurons cause eligibility traces that propagate forward over longer time spans, and are therefore able to coincide with later arising instantaneous error signals (see Fig. 3b).

The theory of e-prop makes a concrete experimentally testable prediction: that the time constant of the eligibility trace for a synapse is correlated with the time constant for the history-dependence of the firing activity of the postsynaptic neuron. It also suggests that the experimentally found diverse time constants of the firing activity of populations of neurons in different brain areas[30] are correlated with their capability to handle corresponding ranges of delays in temporal credit assignment for learning.

Finally, e-prop theory provides a hypothesis for the functional role of the experimentally found diversity of dopamine signals to different populations of neurons[14]. Whereas previous theories of reward-based learning required that the same learning signal is sent to all neurons, the basic Eq. (1) for e-prop postulates that ideal top–down learning signals to a population of neurons depend on its impact on the network performance (loss function), and should therefore be target-specific (see Fig. 2c and Supplementary Note 6). In fact, the learning-to-learn result for e-prop in ref. [31] suggests that prior knowledge about the possible range of learning tasks for a brain area could optimize top–down learning signals even further on an evolutionary time scale, thereby enabling for example learning from few or even a single trial.

Previous methods for training RSNNs did not aim at approximating BPTT. Instead, some of them were relying on control theory to train a chaotic reservoir of spiking neurons[32–34]. Others used the FORCE algorithm[35,36] or variants of it[35,37–39]. However, the FORCE algorithm was not argued to be biologically realistic, as the plasticity rule for each synaptic weight requires knowledge of the current values of all other synaptic weights. The generic task considered in ref. [35] was to learn with supervision how to generate patterns. We show in Supplementary Figs. 1 and 7 that RSNNs can learn such tasks also with a biologically plausible learning method e-prop.

Several methods for approximating stochastic gradient descent in feedforward networks of spiking neurons have been proposed,

see e.g., refs. [40–44]. These employ—like e-prop—a pseudo-gradient to overcome the non-differentiability of a spiking neuron, as proposed previously in refs. [45,46]. References [40,42,43] arrive at a synaptic plasticity rule for feedforward networks that consists—like e-prop—of the product of a learning signal and a derivative (eligibility trace) that describes the dependence of a spike of a neuron $j$ on the weight of an afferent synapse $W_{ji}$. But in a recurrent network, the spike output of $j$ depends on $W_{ji}$ also indirectly, via loops in the network that allow that a spike of neuron $j$ contributes to the firing of other neurons, which in turn affect the firing of the presynaptic neuron $i$. Hence, the corresponding eligibility trace can no longer be locally computed if one transfers these methods for feedforward networks to recurrently connected networks. Therefore, ref. [40] suggests the need to investigate extensions of their approach to RSNNs.

Previous work on the design of online gradient descent learning algorithms for non-spiking RNNs was based on real-time recurrent learning (RTRL)[8]. RTRL itself has rarely been used as its computational complexity per time step is $\mathcal{O}(n^4)$, if $n$ is the number of neurons. But interesting approximations to RTRL have subsequently been proposed (see ref. [47] for a review): some stochastic approximations[48], which are $\mathcal{O}(n^3)$ or only applicable for small networks[49], and also recently two deterministic $\mathcal{O}(n^2)$ approximations[50,51]. The latter were in fact written at the same time as the first publication of e-prop[31]. A structural difference between this paper and[50] is that their approach requires that learning signals are transmitted between the neurons in the RNN, with separately learnt weights.[51] derived for rate based neurons a learning rule similar to random e-prop. But this work did not address other forms of learning than supervised regression, such as RL, nor learning in networks of spiking neurons, or in more powerful types of RNNs with slow hidden variables such as LSTM networks or LSNNs.

E-prop also has complexity $\mathcal{O}(n^2)$, in fact $\mathcal{O}(S)$ if $S$ is the number of synaptic connections. This bound is optimal—except for the constant factor—since this is the asymptotic complexity of just simulating the RNN. The key point of e-prop is that the general form (13) of its eligibility trace collects all contributions to the loss gradient that can be locally computed in a feedforward manner. This general form enables applications to spiking neurons with slowly varying hidden variables, such as neurons with firing rate adaptation, which are essential ingredients of RSNNs to reach the computational power of LSTM networks[3]. We believe that this approach can be extended in future work with a suitable choice of pseudo derivatives to a wide range of biologically more realistic neuron models. It also enables the combination of these rigorously derived eligibility traces with—semantically identical but algorithmically very different—eligibility traces from RL for reward-based e-prop (Eq. (5)), thereby bringing the power of deep RL to RSNNs. As a result, we were able to show in Figs. 2–5 that RSNNs can learn with the biologically plausible rules for synaptic plasticity that arise from the e-prop theory to solve tasks such as phoneme recognition, integrating evidence over time and waiting for the right moment to act, and winning Atari games. These are tasks that are fundamental for modern learning-based AI, but have so far not been solved with RSNNs. Hence, e-prop provides a new perspective of the major open question how intelligent behavior can be learnt and controlled by neural networks of the brain.

Apart from obvious consequences of e-prop for research in neuroscience and cognitive science, e-prop also provides an interesting new tool for approaches in machine learning where BPTT is replaced by approximations in order to improve computational efficiency. We have already shown in Supplementary Fig. 4 that e-prop provides a powerful online learning method for LSTM networks. Furthermore, the combination of eligibility

traces from e-prop with synthetic gradients from ref. [52] even improves performance of LSTM networks for difficult machine learning problems such as the copy-repeat task and the Penn Treebank word prediction task[31]. Other future extensions of e-prop could explore a combination with attention-based models in order to cover multiple timescales.

Finally, e-prop suggests a promising new approach for realizing powerful on-chip learning of RSNNs on neuromorphic chips. Whereas, BPTT is not within reach of current neuromorphic hardware, an implementation of e-prop appears to offer no serious hurdle. Our results show that an implementation of e-prop will provide a qualitative jump in on-chip learning capabilities of neuromorphic hardware.

## Methods

**Network models.** To exhibit the generality of the e-prop approach, we define the dynamics of recurrent neural networks using a general formalism that is applicable to many recurrent neural network models, not only to RSNNs and LSNNs. Also non-spiking models such as LSTM networks fit under this formalism (see Supplementary Note 4). The network dynamics is summarized by the computational graph in Fig. 6. Denoting the observable states (e.g., spikes) as $\mathbf{z}^t$, the hidden states as $\mathbf{h}^t$, the inputs as $\mathbf{x}^t$ and using $\mathbf{W}_j$ to gather all weights of synapses arriving at neuron $j$, the function $M$ defines the update of the hidden state of a neuron $j$: $\mathbf{h}_j^t = M(\mathbf{h}_j^{t-1}, \mathbf{z}^{t-1}, \mathbf{x}^t, \mathbf{W}_j)$ and the function $f$ defines the update of the observable state of a neuron $j$: $z_j^t = f(\mathbf{h}_j^t, \mathbf{z}^{t-1}, \mathbf{x}^t, \mathbf{W}_j)$ ($f$ simplifies to $z_j^t = f(\mathbf{h}_j^t)$ for LIF and ALIF neurons). We chose a discrete time step of $\delta t = 1$ ms for all our simulations. Control experiments with smaller time steps for the task of Fig. 3, reported in Supplementary Fig. 6, suggest that the size of the time step has no significant impact on the performance of e-prop.

**LIF neurons.** Each LIF neuron has a one-dimensional internal state—or hidden variable—$h_j^t$ that consists only of the membrane potential $v_j^t$. The observable state $z_j^t \in \{0, 1\}$ is binary, indicating a spike ($z_j^t = 1$) or no spike ($z_j^t = 0$) at time $t$. The dynamics of the LIF model is defined by the equations:

$$v_j^{t+1} = \alpha v_j^t + \sum_{i \neq j} W_{ji}^{\mathrm{rec}} z_i^t + \sum_i W_{ji}^{\mathrm{in}} x_i^{t+1} - z_j^t v_{\mathrm{th}} \tag{6}$$

$$z_j^t = H\left(v_j^t - v_{\mathrm{th}}\right). \tag{7}$$

$W_{ji}^{\mathrm{rec}}$ ($W_{ji}^{\mathrm{in}}$) is the synaptic weight from network (input) neuron $i$ to neuron $j$. The decay factor $\alpha$ in (6) is given by $e^{-\delta t/\tau_m}$, where $\tau_m$ (typically 20 ms) is the membrane time constant. $\delta t$ denotes the discrete time step size, which is set to 1 ms in our simulations. $H$ denotes the Heaviside step function. Note that we deleted in Eq. (6) the factor $1 - \alpha$ that occurred in the corresponding equation (4) in the supplement of ref. [3]. This simplifies the notation in our derivations, and has no impact on the model if parameters like the threshold voltage are scaled accordingly.

Owing to the term $-z_j^t v_{\mathrm{th}}$ in Eq. (6), the neurons membrane potential is reduced by a constant value after an output spike, which relates our model to the spike response model[53]. To introduce a simple model of neuronal refractoriness, we further assume that $z_j^t$ is fixed to 0 after each spike of neuron $j$ for a short refractory period of 2–5 ms depending on the simulation.

**LSNNs.** According to the database of the Allen Institute[2], a fraction of neurons between ~20% (in mouse visual cortex) and 40% (in the human frontal lobe) exhibit SFA. It had been shown in ref. [3] that the inclusion of neuron models with SFA—via a time-varying firing threshold as slow hidden variable—drastically enhances computing capabilities of RSNN models. Hence, we consider here the same simple model for neurons with SFA as in ref. [3], to which we refer as ALIF neuron. This model is basically the same as the GLIF$_2$ model in the Technical White Paper on generalized LIF (GLIF) models from ref. [2]. LSNNs are recurrently connected networks that consist of LIF and ALIF neurons. ALIF neurons $j$ have a second hidden variable $a_j^t$, which denotes the variable component of its firing threshold. As a result, their internal state is a two-dimensional vector $\mathbf{h}_j^t \overset{\mathrm{def}}{=} [v_j^t, a_j^t]$. Their threshold potential $A_j^t$ increases with every output spike and decreases exponentially back to the baseline threshold $v_{\mathrm{th}}$. This can be described by

$$A_j^t = v_{\mathrm{th}} + \beta a_j^t, \tag{8}$$

$$z_j^t = H(v_j^t - A_j^t), \tag{9}$$

with a threshold adaptation according to

$$a_j^{t+1} = \rho a_j^t + z_j^t, \tag{10}$$

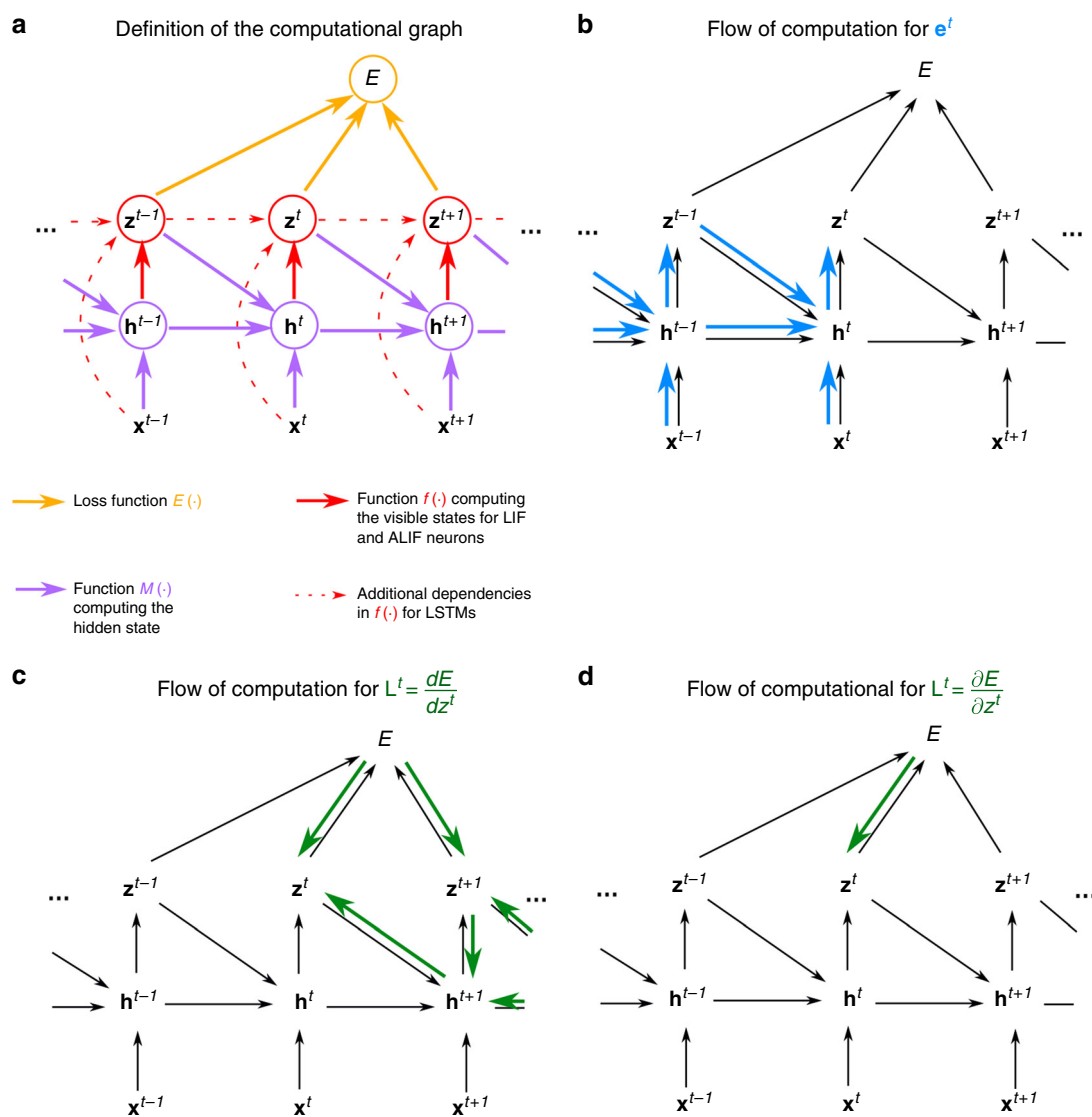

**Fig. 6 Computational graph and gradient propagations. a** Assumed mathematical dependencies between hidden neuron states $\mathbf{h}_j^t$, neuron outputs $\mathbf{z}^t$, network inputs $\mathbf{x}^t$, and the loss function $E$ through the mathematical functions $E(\cdot)$, $M(\cdot)$, $f(\cdot)$ are represented by colored arrows. **b–d** The flow of computation for the two components $\mathbf{e}^t$ and $\mathbf{L}^t$ that merge into the loss gradients of Eq. (3) can be represented in similar graphs. **b** Following Eq. (14), the flow of the computation of the eligibility traces $e_{ji}^t$ is going forward in time. **c** Instead, the ideal learning signals $L_j^t = \frac{dE}{dz_j^t}$ requires to propagate gradients backward in time. **d** Hence, while $e_{ji}^t$ is computed exactly, $L_j^t$ is approximated in e-prop applications to yield an online learning algorithm.

where the decay factor $\rho$ is given by $e^{-\delta t / \tau_a}$, and $\tau_a$ is the adaptation time constant that is typically chosen to be on the same time scale as the length of the working memory that is a relevant for a given task. The effects of SFA can last for several seconds in neocortical neurons, in factor up to 20 s according to experimental data[54]. We refer to a RSNN as LSNN if some of its neurons are adaptive. We chose a fraction between 25 and 40% of the neurons to be adapting.

In relation to the more general formalism represented in the computational graph in Fig. 6, Eqs. (6) and (10) define $M(\mathbf{h}_j^{t-1}, \mathbf{z}^{t-1}, \mathbf{x}^t, \mathbf{W}_j)$, and Eqs. (7) and (9) define $f(\mathbf{h}_j^t)$.

**Gradient descent for RSNNs.** Gradient descent is problematic for spiking neurons because of the step function $H$ in Eq. (7). We overcome this issue as in refs. [3,55]: the non-existing derivative $\frac{\partial z_j^t}{\partial v_j^t}$ is replaced in simulations by a simple nonlinear function of the membrane potential that is called the pseudo derivative. Outside of the refractory period, we choose a pseudo derivative of the form $\psi_j^t = \frac{1}{v_{th}} \gamma_{pd} \max\left(0, 1 - \left|\frac{v_j^t - A_j^t}{v_{th}}\right|\right)$ where $\gamma_{pd} = 0.3$ for ALIF neurons, and for LIF neurons $A_j^t$ is replaced by $v_{th}$. During the refractory period the pseudo derivative is set to 0.

**Network output and loss functions.** We assume that network outputs $y_k^t$ are real-valued and produced by leaky output neurons (readouts) $k$, which are not recurrently connected:

$$y_k^t = \kappa y_k^{t-1} + \sum_j W_{kj}^{\text{out}} z_j^t + b_k^{\text{out}}, \tag{11}$$

where $\kappa \in [0, 1]$ defines the leak and $b_k^{\text{out}}$ denotes the output bias. The leak factor $\kappa$ is given for spiking neurons by $e^{-\delta t / \tau_{\text{out}}}$, where $\tau_{\text{out}}$ is the membrane time constant. Note that for non-spiking neural networks (such as for LSTM networks), temporal smoothing of the network observable state is not necessary. In this case, one can use $\kappa = 0$.

The loss function $E(\mathbf{z}^1, \ldots, \mathbf{z}^T)$ quantifies the network performance. We assume that it depends only on the observable states $\mathbf{z}^1, \ldots, \mathbf{z}^T$ of the network neurons. For instance, for a regression problem we define $E$ as the mean square error $E = \frac{1}{2} \sum_{t,k} (y_k^t - y_k^{*,t})^2$ between the network outputs $y_k^t$ and target values $y_k^{*,t}$. For classification or RL tasks the loss function, $E$ has to be re-defined accordingly.

**Notation for derivatives.** We distinguish the total derivative $\frac{dE}{dz^t}(\mathbf{z}^1, \ldots, \mathbf{z}^T)$, which takes into account how $E$ depends on $\mathbf{z}^t$ also indirectly through influence of $\mathbf{z}^t$ on the other variables $\mathbf{z}^{t+1}, \ldots, \mathbf{z}^T$, and the partial derivative $\frac{\partial E}{\partial z^t}(\mathbf{z}^1, \ldots, \mathbf{z}^T)$, which quantifies only the direct dependence of $E$ on $\mathbf{z}^t$.

Analogously for the hidden state $\mathbf{h}_j^t = M(\mathbf{h}_j^{t-1}, \mathbf{z}^{t-1}, \mathbf{x}^t, \mathbf{W}_j)$, the partial derivative $\frac{\partial M}{\partial \mathbf{h}_j^{t-1}}$ denotes the partial derivative of $M$ with respect to $\mathbf{h}_j^{t-1}$. It only

quantifies the direct influence of $\mathbf{h}_j^{t-1}$ on $\mathbf{h}_j^t$ and it does not take into account how $\mathbf{h}_j^{t-1}$ indirectly influences $\mathbf{h}_j^t$ via the observable states $\mathbf{z}^{t-1}$. To improve readability, we also use the following abbreviations: $\frac{\partial \mathbf{h}_j^t}{\partial \mathbf{h}_j^{t-1}} \overset{\text{def}}{=} \frac{\partial M}{\partial \mathbf{h}_j^{t-1}}(\mathbf{h}_j^{t-1}, \mathbf{z}^{t-1}, \mathbf{x}^t, \mathbf{W}_j)$, $\frac{\partial \mathbf{h}_j^t}{\partial W_{ji}} \overset{\text{def}}{=} \frac{\partial M}{\partial W_{ji}}(\mathbf{h}_j^{t-1}, \mathbf{z}^{t-1}, \mathbf{x}^t, \mathbf{W}_j)$, and $\frac{\partial z_j^t}{\partial \mathbf{h}_j^t} \overset{\text{def}}{=} \frac{\partial f}{\partial h^t}(\mathbf{h}_j^t, \mathbf{z}^{t-1}, \mathbf{x}^t, \mathbf{W}_j)$.

**Notation for temporal filters.** For ease of notation, we use the operator $\mathcal{F}_\alpha$ to denote the low-pass filter such that, for any time series $x_t$:

$$\mathcal{F}_\alpha(x^t) = \alpha \mathcal{F}_\alpha(x^{t-1}) + x^t, \tag{12}$$

and $\mathcal{F}_\alpha(x^0) = x^0$. In the specific case of the time series $z_j^t$ and $e_{ji}^t$, we simplify notation further and write $\bar{z}_j^t$ and $\bar{e}_{ji}^t$ for $\mathcal{F}_\alpha(z_j)^t$ and $\mathcal{F}_\kappa(e_{ji})^t$.

**Mathematical basis for e-prop.** We provide here the proof of the fundamental Eq. (1) for e-prop (for the case where the learning signal $L_j^t$ has its ideal value $\frac{dE}{dz_j^t}$)

$$\frac{dE}{dW_{ji}} = \sum_t \frac{dE}{dz_j^t} \cdot \left[ \frac{dz_j^t}{dW_{ji}} \right]_{\text{local}},$$

with the eligibility trace

$$e_{ji}^t \overset{\text{def}}{=} \left[ \frac{dz_j^t}{dW_{ji}} \right]_{\text{local}} \overset{\text{def}}{=} \frac{\partial z_j^t}{\partial \mathbf{h}_j^t} \underbrace{\sum_{t' \leq t} \frac{\partial \mathbf{h}_j^t}{\partial \mathbf{h}_j^{t-1}} \cdots \frac{\partial \mathbf{h}_j^{t'+1}}{\partial \mathbf{h}_j^{t'}} \cdot \frac{\partial \mathbf{h}_j^{t'}}{\partial W_{ji}}}_{\overset{\text{def}}{=} \epsilon_{ji}^t}. \tag{13}$$

The second factor $\epsilon_{ji}^t$, which we call eligibility vector, obviously satisfies the recursive equation

$$\epsilon_{ji}^t = \frac{\partial \mathbf{h}_j^t}{\partial \mathbf{h}_j^{t-1}} \cdot \epsilon_{ji}^{t-1} + \frac{\partial \mathbf{h}_j^t}{\partial W_{ji}}, \tag{14}$$

where $\cdot$ denotes the dot product. This provides the rule for the online computation of $\epsilon_{ji}^t$, and hence of $e_{ji}^t = \frac{\partial z_j^t}{\partial \mathbf{h}_j^t} \cdot \epsilon_{ji}^t$.

We start from a classical factorization of the loss gradients in recurrent neural networks that arises for instance in equation (12) of ref. [56] to describe BPTT. This classical factorization can be justified by unrolling an RNN into a large feedforward network, where each layer ($l$) represents one time step. In a feedforward network, the loss gradients with respect to the weights $W_{ji}^{(l)}$ of layer ($l$) are given by $\frac{dE}{dW_{ji}^{(l)}} = \frac{dE}{dh_j^{(l)}} \frac{\partial h_j^{(l)}}{\partial W_{ji}^{(l)}}$. But as the weights are shared across the layers when representing a recurrent network, the summation of these gradients over the layers $l$ of the unrolled RNN yields this classical factorization of the loss gradients:

$$\frac{dE}{dW_{ji}} = \sum_{t'} \frac{dE}{d\mathbf{h}_j^{t'}} \cdot \frac{\partial \mathbf{h}_j^{t'}}{\partial W_{ji}}. \tag{15}$$

Note that the first factor $\frac{dE}{dh_j^{t'}}$ in these products also needs to take into account how the internal state $\mathbf{h}_j$ of neuron $j$ evolves during subsequent time steps, and whether it influences firing of $j$ at later time steps. This is especially relevant for ALIF neurons and other biologically realistic neuron models with slowly changing internal states. Note that this first factor of (15) is replaced in the e-prop equation (13) by the derivative $\frac{dE}{dz_j^{t'}}$ of $E$ with regard to the observable variable $z_j^{t'}$. There, the evolution of the internal state of neuron $j$ is pushed into the second factor, the eligibility trace $e_{ji}$, which collects in e-prop all online computable factors of the loss gradient that just involve neurons $j$ and $i$.

Now we show that one can re-factorize the expression (15) and prove that the loss gradients can also be computed using the new factorization (13) that underlies e-prop. In the steps of the subsequent proof until Eq. (19), we decompose the term $\frac{dE}{d\mathbf{h}_j^{t'}}$ into a series of learning signals $L_j^t = \frac{dE}{dz_j^t}$ and local factors $\frac{\partial \mathbf{h}_j^t}{\partial \mathbf{h}_j^{t'}}$ for $t \geq t'$. Those local factors will later be used to transform the partial derivative $\frac{\partial \mathbf{h}_j^{t'}}{\partial W_{ji}}$ from Eq. (15) into the eligibility vector $\epsilon_{ji}^t$ that integrates the whole history of the synapse up to time $t$, not just a single time step. To do so, we express $\frac{dE}{d\mathbf{h}_j^{t'}}$ recursively as a function of the same derivative at the next time step $\frac{dE}{d\mathbf{h}_j^{t'+1}}$ by applying the chain rule at the node $\mathbf{h}_j^t$ for $t = t'$ of the computational graph shown in Fig. 6c:

$$\frac{dE}{d\mathbf{h}_j^{t'}} = \frac{dE}{dz_j^{t'}} \frac{\partial z_j^{t'}}{\partial \mathbf{h}_j^{t'}} + \frac{dE}{d\mathbf{h}_j^{t'+1}} \frac{\partial \mathbf{h}_j^{t'+1}}{\partial \mathbf{h}_j^{t'}} \tag{16}$$

$$= L_j^{t'} \frac{\partial z_j^{t'}}{\partial \mathbf{h}_j^{t'}} + \frac{dE}{d\mathbf{h}_j^{t'+1}} \frac{\partial \mathbf{h}_j^{t'+1}}{\partial \mathbf{h}_j^{t'}}, \tag{17}$$

where we defined the learning signal $L_j^{t'}$ as $\frac{dE}{dz_j^{t'}}$. The resulting recursive expansion

ends at the last time step $T$ of the computation of the RNN, i.e., $\frac{dE}{d\mathbf{h}_j^{T+1}} = 0$. If one repeatedly substitutes the recursive formula (17) into the classical factorization (15) of the loss gradients, one gets:

$$\frac{dE}{dW_{ji}} = \sum_{t'} \left( L_j^{t'} \frac{\partial z_j^{t'}}{\partial \mathbf{h}_j^{t'}} + \frac{dE}{d\mathbf{h}_j^{t'+1}} \frac{\partial \mathbf{h}_j^{t'+1}}{\partial \mathbf{h}_j^{t'}} \right) \cdot \frac{\partial \mathbf{h}_j^{t'}}{\partial W_{ji}} \tag{18}$$

$$= \sum_{t'} \left( L_j^{t'} \frac{\partial z_j^{t'}}{\partial \mathbf{h}_j^{t'}} + \left( L_j^{t'+1} \frac{\partial z_j^{t'+1}}{\partial \mathbf{h}_j^{t'+1}} + (\cdots) \frac{\partial \mathbf{h}_j^{t'+2}}{\partial \mathbf{h}_j^{t'+1}} \right) \frac{\partial \mathbf{h}_j^{t'+1}}{\partial \mathbf{h}_j^{t'}} \right) \cdot \frac{\partial \mathbf{h}_j^{t'}}{\partial W_{ji}}. \tag{19}$$

The following equation is the main equation for understanding the transformation from BPTT into e-prop. The key idea is to collect all terms $\frac{\partial \mathbf{h}_j^{t'+1}}{\partial \mathbf{h}_j^{t'}}$, which are multiplied with the learning signal $L_j^t$ at a given time $t$. These are only terms that concern events in the computation of neuron $j$ up to time $t$, and they do not depend on other future losses or variable values. To this end, we write the term in parentheses in Eq. (19) into a second sum indexed by $t$ and exchange the summation indices to pull out the learning signal $L_j^t$. This expresses the loss gradient of $E$ as a sum of learning signals $L_j^t$ multiplied by some factor indexed by $ji$, which we define as the eligibility trace $e_{ji}^t \in \mathbb{R}$. The main factor of it is the eligibility vector $\epsilon_{ji}^t \in \mathbb{R}^d$, which has the same dimension as the hidden state $\mathbf{h}_j^t$:

$$\frac{dE}{dW_{ji}} = \sum_{t'} \sum_{t \geq t'} L_j^t \frac{\partial z_j^t}{\partial \mathbf{h}_j^t} \frac{\partial \mathbf{h}_j^t}{\partial \mathbf{h}_j^{t-1}} \cdots \frac{\partial \mathbf{h}_j^{t'+1}}{\partial \mathbf{h}_j^{t'}} \cdot \frac{\partial \mathbf{h}_j^{t'}}{\partial W_{ji}} \tag{20}$$

$$= \sum_t L_j^t \frac{\partial z_j^t}{\partial \mathbf{h}_j^t} \underbrace{\sum_{t' \leq t} \frac{\partial \mathbf{h}_j^t}{\partial \mathbf{h}_j^{t-1}} \cdots \frac{\partial \mathbf{h}_j^{t'+1}}{\partial \mathbf{h}_j^{t'}} \cdot \frac{\partial \mathbf{h}_j^{t'}}{\partial W_{ji}}}_{\overset{\text{def}}{=} \epsilon_{ji}^t}. \tag{21}$$

This completes the proof of Eqs. (1), (3), (13).

**Derivation of eligibility traces LIF neurons.** The eligibility traces for LSTMs are derived in the supplementary materials. Below we provide the derivation of eligibility traces for spiking neurons.

We compute the eligibility trace of a synapse of a LIF neuron without adaptive threshold (Eq. (6)). Here, the hidden state $\mathbf{h}_j^t$ of a neuron consists just of the membrane potential $v_j^t$ and we have $\frac{\partial \mathbf{h}_j^{t+1}}{\partial \mathbf{h}_j^t} = \frac{\partial v_j^{t+1}}{\partial v_j^t} = \alpha$ and $\frac{\partial v_j^t}{\partial W_{ji}} = z_i^{t-1}$ (for a derivation of the eligibility traces taking the reset into account we refer to Supplementary Note 1). Using these derivatives and Eq. (14), one obtains that the eligibility vector is the low-pass filtered presynaptic spike-train,

$$\epsilon_{ji}^{t+1} = \mathcal{F}_\alpha(z_i^t) \overset{\text{def}}{=} \bar{z}_i^t, \tag{22}$$

and following Eq. (13), the eligibility trace is:

$$e_{ji}^{t+1} = \psi_j^{t+1} \bar{z}_i^t. \tag{23}$$

For all neurons, $j$ the derivations in the next sections also hold for synaptic connections from input neurons $i$, but one needs to replace the network spikes $z_i^{t-1}$ by the input spikes $x_i^t$ (the time index switches from $t-1$ to $t$ because the hidden state $\mathbf{h}_j^t = M(\mathbf{h}_j^{t-1}, \mathbf{z}^{t-1}, \mathbf{x}^t, \mathbf{W}_j)$ is defined as a function of the input at time $t$ but the preceding recurrent activity). For simplicity, we have focused on the case where transmission delays between neurons in the RSNN are just 1 ms. If one uses more realistic length of delays $d$, this $d$ appears in Eqs. (23)–(25) instead of $-1$ as the most relevant time point for presynaptic firing (see Supplementary Note 1). This moves resulting synaptic plasticity rules closer to experimentally observed forms of STDP.

**Eligibility traces for ALIF neurons.** The hidden state of an ALIF neuron is a two dimensional vector $\mathbf{h}_j^t = [v_j^t, a_j^t]$. Hence, a two-dimensional eligibility vector $\epsilon_{ji}^t \overset{\text{def}}{=} [\epsilon_{ji,v}^t, \epsilon_{ji,a}^t]$ is associated with the synapse from neuron $i$ to neuron $j$, and the matrix $\frac{\partial \mathbf{h}_j^{t+1}}{\partial \mathbf{h}_j^t}$ is a $2 \times 2$ matrix. The derivatives $\frac{\partial a_j^{t+1}}{\partial a_j^t}$ and $\frac{\partial a_j^{t+1}}{\partial v_j^t}$ capture the dynamics of the adaptive threshold. Hence, to derive the computation of eligibility traces, we substitute the spike $z_j$ in Eq. (10) by its definition given in Eq. (9). With this convention, one finds that the diagonal of the matrix $\frac{\partial \mathbf{h}_j^{t+1}}{\partial \mathbf{h}_j^t}$ is formed by the terms $\frac{\partial v_j^{t+1}}{\partial v_j^t} = \alpha$ and $\frac{\partial a_j^{t+1}}{\partial a_j^t} = \rho - \psi_j^t \beta$. Above and below the diagonal, one finds, respectively, $\frac{\partial v_j^{t+1}}{\partial a_j^t} = 0$, $\frac{\partial a_j^{t+1}}{\partial v_j^t} = \psi_j^t$. Seeing that $\frac{\partial \mathbf{h}_j^t}{\partial W_{ji}} = \left[ \frac{\partial v_j^t}{\partial W_{ji}}, \frac{\partial a_j^t}{\partial W_{ji}} \right] = [z_i^{t-1}, 0]$, one can finally compute the eligibility traces using Eq. (13). The component of the eligibility vector associated with the membrane potential remains the same as in the LIF case and only depends on the presynaptic neuron: $\epsilon_{ji,v}^t = \bar{z}_i^{t-1}$. For the component

associated with the adaptive threshold we find the following recursive update:

$$\epsilon_{ji,a}^{t+1} = \psi_j^t \bar{z}_i^{t-1} + (\rho - \psi_j^t \beta) \epsilon_{ji,a}^t , \tag{24}$$

and, since $\frac{\partial z_j^t}{\partial \mathbf{h}_j^t} = \left[ \frac{\partial z_j^t}{\partial v_j^t}, \frac{\partial z_j^t}{\partial a_j^t} \right] = \left[ \psi_j^t, -\beta \psi_j^t \right]$, this results in an eligibility trace of the form:

$$e_{ji}^t = \psi_j^t \left( \bar{z}_i^{t-1} - \beta \epsilon_{ji,a}^t \right). \tag{25}$$

Recall that the constant $\rho = \exp(-\frac{\delta t}{\tau_a})$ arises from the adaptation time constant $\tau_a$, which typically lies in the range of hundreds of milliseconds to a few seconds in our experiments, yielding values of $\rho$ between 0.995 and 0.9995. The constant $\beta$ is typically of the order of 0.07 in our experiments.

To provide a more interpretable form of eligibility trace that fits into the standard form of local terms considered in three-factor learning rules[12], one may drop the term $-\psi_j^t \beta$ in Eq. (24). This approximation $\hat{\epsilon}_{ji,a}^t$ of Eq. (24) becomes an exponential trace of the post-pre pairings accumulated within a time window as large as the adaptation time constant:

$$\hat{\epsilon}_{ji,a}^{t+1} = \mathcal{F}_\rho \left( \psi_j^t \bar{z}_i^{t-1} \right). \tag{26}$$

The eligibility traces are computed with Eq. (24) in most experiments, but the performances obtained with symmetric e-prop and this simplification were indistinguishable in the task where temporal credit assignment is difficult of Fig. 3.

**Synaptic plasticity rules resulting from e-prop.** An exact computation of the ideal learning signal $\frac{dE}{dz_j^t}$ in Eq. (1) requires to back-propagate gradients through time (see Fig. 6c). For online e-prop, we replace it with the partial derivative $\frac{\partial E}{\partial z_j^t}$, which can be computed online. Implementing the weight updates with gradient descent and learning rate $\eta$, all the following plasticity rules are derived from the formula

$$\Delta W_{ji}^{\text{rec}} = -\eta \sum_t \frac{\partial E}{\partial z_j^t} e_{ji}^t . \tag{27}$$

Note that in the absence of the superscript $t$, $\Delta W_{ji}$ denotes the cumulated weight change over one trial or batch of consecutive trials but not the instantaneous weight update. This can be implemented online by accumulating weight updates in a hidden synaptic variable. Note also that the weight updates derived in the following for the recurrent weights $W_{ji}^{\text{rec}}$ also apply to the inputs weights $W_{ji}^{\text{in}}$. For the output weights and biases, the derivation does not require the theory of e-prop, and the weight updates can be found in Supplementary Note 3.

In the case of a regression problem with targets $y_k^{*,t}$ and outputs $y_k^t$ defined in Eq. (11), we define the loss function $E = \frac{1}{2} \sum_{t,k} (y_k^t - y_k^{*,t})^2$. This results in a partial derivative of the form $\frac{\partial E}{\partial z_j^t} = \sum_k W_{kj}^{\text{out}} \sum_{t' \geq t} (y_k^{t'} - y_k^{*,t'}) \kappa^{t'-t}$. This seemingly provides an obstacle for online learning, because the partial derivative is a weighted sum over future errors. But this problem can be resolved as one can interchange the two summation indices in the expression for the weight updates (see Supplementary Note 3). In this way, the sum over future events transforms into a low-pass filtering of the eligibility traces $\bar{e}_{ji}^t = \mathcal{F}_\kappa(e_{ji}^t)$, and the resulting weight update can be written as

$$\Delta W_{ji}^{\text{rec}} = -\eta \sum_t \underbrace{\left( \sum_k B_{jk}(y_k^t - y_k^{*,t}) \right)}_{=L_j^t} \bar{e}_{ji}^t . \tag{28}$$

For classification tasks, we assume that $K$ target categories are provided in the form of a one-hot encoded vector $\pi^{*,t}$ with $K$ dimensions. We define the probability for class $k$ predicted by the network as $\pi_k^t = \text{softmax}_k(y_1^t, \dots, y_K^t) = \exp(y_k^t) / \sum_{k'} \exp(y_{k'}^t)$, and the loss function for classification tasks as the cross-entropy error $E = -\sum_{t,k} \pi_k^{*,t} \log \pi_k^t$. The plasticity rule resulting from e-prop reads (see derivation in Supplementary Note 3):

$$\Delta W_{ji}^{\text{rec}} = -\eta \sum_t \underbrace{\left( \sum_k B_{jk}(\pi_k^t - \pi_k^{*,t}) \right)}_{=L_j^t} \bar{e}_{ji}^t . \tag{29}$$

**Reward-based e-prop: application of e-prop to deep RL.** For RL, the network interacts with an external environment. At any time, $t$ the environment can provide a positive or negative reward $r^t$. Based on the observations $\mathbf{x}^t$ that are perceived, the network has to commit to actions $a^{t_0}, \dots, a^{t_n}, \dots$ at certain decision times $t_0, \dots, t_n, \dots$. Each action $a^t$ is sampled from a probability distribution $\pi(\cdot | \mathbf{y}^t)$, which is also referred to as the policy of the RL agent. The policy is defined as function of the network outputs $\mathbf{y}^t$, and is chosen here to be a categorical distribution of $K$ discrete action choices. We assume that the agent chooses action $k$ with probability $\pi_k^t = \pi(a^t = k | \mathbf{y}^t) = \text{softmax}_k(y_1^t, \dots, y_K^t) = \exp(y_k^t) / \sum_{k'} \exp(y_{k'}^t)$.

The goal of RL is to maximize the expected sum of discounted rewards. That is, we want to maximize the expected return at time $t = 0$, $\mathbb{E}[R^0]$, where the return at

time $t$ is defined as $R^t = \sum_{t' \geq t} \gamma^{t'-t} r^{t'}$ with a discount factor $\gamma \leq 1$. The expectation is taken over the agent actions $a^t$, the rewards $r^t$ and the observations from the environment $\mathbf{x}^t$. We approach this optimization problem by using the actor-critic variant of the policy gradient algorithm, which applies gradient ascent to maximize $\mathbb{E}[R^0]$. The basis of the estimated gradient relies on an estimation of the policy gradient, as shown in Section 13.3 in ref. [27]. There, the resulting weight update is given in equation (13.8) where $G_t$ refers to the return $R^t$. Similarly, the gradient $\frac{d\mathbb{E}[R^0]}{dW_{ji}}$ is proportional to $\mathbb{E}\left[ \sum_{t_n} R^{t_n} \frac{d \log \pi(a^{t_n}|\mathbf{y}^{t_n})}{dW_{ji}} \right]$, which is easier to compute because the expectation can be estimated by an average over one or many trials. Following this strategy, we define the per-trial loss function $E_\pi$ as a function of the sequence of actions $a^{t_0}, \dots, a^{t_n}, \dots$ and rewards $r^0, \dots, r^T$ sampled during this trial:

$$E_\pi(\mathbf{z}^0, \dots, \mathbf{z}^T, a^{t_0}, \dots a^{t_n}, \dots, r^0, \dots, r^T) \overset{\text{def}}{=} -\sum_n R^{t_n} \log \pi(a^{t_n}|\mathbf{y}^{t_n}) . \tag{30}$$

And thus:

$$\frac{d\mathbb{E}[R^0]}{dW_{ji}} \propto \mathbb{E}\left[ \sum_{t_n} R^{t_n} \frac{d \log \pi(a^{t_n}|\mathbf{y}^{t_n})}{dW_{ji}} \right] = -\mathbb{E}\left[ \frac{dE_\pi}{dW_{ji}} \right] . \tag{31}$$

Intuitively, given a trial with high rewards, policy gradient changes the network output $\mathbf{y}$ to increase the probability of the actions $a^{t_n}$ that occurred during this trial. In practice, the gradient $\frac{dE_\pi}{dW_{ji}}$ is known to have high variance and the efficiency of the learning algorithm can be improved using the actor-critic variant of the policy gradient algorithm. It involves the policy $\pi$ (the actor) and an additional output neuron $V^t$, which predicts the value function $\mathbb{E}[R^t]$ (the critic). The actor and the critic are learnt simultaneously by defining the loss function as

$$E = E_\pi + c_V E_V , \tag{32}$$

where $E_\pi = -\sum_n R^{t_n} \log \pi(a^{t_n}|\mathbf{y}^{t_n})$ measures the performance of the stochastic policy $\pi$, and $E_V = \sum_t \frac{1}{2}(R^t - V^t)^2$ measures the accuracy of the value estimate $V^t$.

As $V^t$ is independent of the action $a^t$ one can show that $0 = \mathbb{E}\left[ V^{t_n} \frac{d \log \pi(a^{t_n}|\mathbf{y}^{t_n})}{dW_{ji}} \right]$. We can use that to define an estimator $\widehat{\frac{dE}{dW_{ji}}}$ of the loss gradient with reduced variance:

$$\mathbb{E}\left[ \frac{dE}{dW_{ji}} \right] = \mathbb{E}\left[ \frac{dE_\pi}{dW_{ji}} \right] + c_V \mathbb{E}\left[ \frac{dE_V}{dW_{ji}} \right] \tag{33}$$

$$= \mathbb{E}\underbrace{\left[ -\sum_{t_n}(R^{t_n} - V^{t_n}) \frac{d \log \pi(a^{t_n}|\mathbf{y}^{t_n})}{dW_{ji}} + c_V \frac{dE_V}{dW_{ji}} \right]}_{\overset{\text{def}}{=} \widehat{\frac{dE}{dW_{ji}}}} , \tag{34}$$

similarly as in equation (13.11) of Section 13.4 in ref. [27]. A difference in notation is that $b(S_t)$ refers to our value estimation $V^t$. In addition, Eq. (34) already includes the gradient $\frac{dE_V}{dW_{ji}}$ that is responsible for learning the value prediction. Until now, this derivation follows the classical definition of the actor-critic variant of policy gradient, and the gradient $\widehat{\frac{dE}{dW_{ji}}}$ can be computed with BPTT. To derive reward-based e-prop, we follow instead the generic online approximation of e-prop as in Eq. (27) and approximate $\widehat{\frac{dE}{dW_{ji}}}$ by a sum of terms of the form $\widehat{\frac{\partial E}{\partial z_j^t}} e_{ji}^t$ with

$$\widehat{\frac{\partial E}{\partial z_j^t}} = -\sum_n (R^{t_n} - V^{t_n}) \frac{\partial \log \pi(a^{t_n}|\mathbf{y}^{t_n})}{\partial z_j^t} + c_V \frac{\partial E_V}{\partial z_j^t} . \tag{35}$$

We choose this estimator $\widehat{\frac{\partial E}{\partial z_j^t}}$ of the loss derivative because it is unbiased and has a low variance, more details are given in Supplementary Note 5. We derive below the resulting synaptic plasticity rule as needed to solve the task of Fig. 4, 5. For the case of a single action as used in Supplementary Fig. 5 we refer to Supplementary Note 5.

When there is a delay between the action and the reward or, even harder, when a sequence of many actions lead together to a delayed reward, the loss function $E$ cannot be computed online because the evaluation of $R^{t_n}$ requires knowledge of future rewards. To overcome this, we introduce temporal difference errors $\delta^t = r^t + \gamma V^{t+1} - V^t$ (see Fig. 4), and use the equivalence between the forward and backward view in RL[27]. Using the one-hot encoded action $\mathbb{1}_{a^t=k}$ at time $t$, which assumes the value 1 if and only if $a^t = k$ (else it has value 0), we arrive at the following synaptic plasticity rules for a general actor-critic algorithm with e-prop (see Supplementary Note 5):

$$\Delta W_{ji}^{\text{rec}} = -\eta \sum_t \delta^t \mathcal{F}_\gamma \left( L_j^t \bar{e}_{ji}^t \right) \quad \text{for} \tag{36}$$

$$L_j^t = -c_V B_j^V + \sum_k B_{jk}(\pi_k^t - \mathbb{1}_{a^t=k}) , \tag{37}$$

where we define the term $\pi_k^t - \mathbb{1}_{a^t=k}$ to have value zero when no action is taken at

time $t$. $B_j^v$ is here the weight from the output neuron for the value function to neuron $j$, and the weights $B_{jk}^\pi$ denote the weights from the outputs for the policy.

A combination of reward prediction error and neuron-specific learning signal was previously used in a plasticity rule for feedforward networks inspired by neuroscience[57,58]. Here, it arises from the approximation of BPTT by e-prop in RSNNs solving RL problems. Note that the filtering $\mathcal{F}_\gamma$ requires an additional eligibility trace per synapse. This arises from the temporal difference learning in RL[27]. It depends on the learning signal and does not have the same function as the eligibility trace $e_{ji}^t$.

## Data availability

The data that support the findings of this study are available from the authors upon reasonable request. Data for the TIMIT and ATARI benchmark tasks were published in previous works[29] with DOI [https://doi.org/10.1613/jair.3912] and ref. [20] with DOI [https://doi.org/10.6028/nist.ir.4930]. Data for the temporal credit assignment task are generated by a custom code provided in the abovementioned code repository.

## Code availability

An implementation of e-prop solving the tasks of Figs. 2–5 is made public together with the publication of this paper https://github.com/IGITUGraz/eligibility_propagation.

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

## Acknowledgements

This research/project was supported by the Human Brain Project (grant agreement number 785907), the SYNCH project (grant agreement number 824162) of the European Union and a grant from Intel. We gratefully acknowledge the support of NVIDIA Corporation with the donation of the Quadro P6000 GPU used for this research. Computations were carried out on the Human Brain Project PCP Pilot Systems at the Juelich Supercomputing Centre, which received co-funding from the European Union (grant agreement number 604102) and on the Vienna Scientific Cluster (VSC). We thank Thomas Bohnstingl, Wulfram Gerstner, Christopher Harvey, Martin Vinck, Jason MacLean, Adam Santoro, Christopher Summerfield, and Yuqing Zhu for helpful comments on an earlier version of the manuscript. Special thanks go to Arjun Rao for letting us use his code for the regularization of membrane voltages.

## Author contributions

G.B., F.S., A.S., and W.M. conceived the work, G.B., F.S., A.S., E.H., and D.S. carried out experiments, G.B., F.S., A.S., E.H., D.S., R.L., and W.M. contributed to the writing of the paper.

## Competing interests

The authors declare no conflict of interest.
