## [Peer Review File · Nature Communications]

Reviewers' comments:

Reviewer #1 (Remarks to the Author):

Review of „A solution to the learning dilemma for recurrent networks of spiking neurons“ by Bellec et al

Data of review: Jan 2, 2020

Competence of this reviewer:

Background in mathematics (dynamical systems) and RNNs, working in theoretical machine learning, only peripherally familiar with computational neuroscience.

Basis of this review:

I read the Introduction, Results, Discussion and the Methods up to (including) „Mathematical basis for e-prop“ in detail and only skimmed the rest due to lack of time. The „Mathematical basis for e-prop“ subsection of „Methods“ contains the core conceptual and mathematical innovation; the remaining parts which I only read superficially can be considered „mechanical“ workouts of the core idea and I trust the authors that this was done correctly.

Summary of contribution:

This work proposes a novel, approximate mathematical formulation of gradient descent optimization of recurrent neural networks (RNNs), called „e-prop“, and demonstrates the functionality of the new approach in nontrivial case studies (phoneme recognition, a maze navigation task with long short-term memory demands, Atari game learning) where it essentially comes out on a par with existing RNN training methods based on backpropagation through time (BPTT) and RNN-enhanced reinforcement learning. The main advantages of the novel formulation are

- generality: the scheme is applicable (with adequate specializations of basic formulas in each case) to many types of RNNs, including rate-based / LSTM networks and spiking variants; and to diverse sorts of loss functions (demonstrated in the paper: regression, classification, reinforcement learning)

- computational efficiency and biological plausibility: the new method obviates the biologically impossible and computationally expensive demands of BPPT for storing and processing many time-slice copies of the RNN. Weight adaptation is based only on information locally available at the concerned synapse, plus reward/teacher feedback information which needs no global integration of information (and is argued by the authors to be compatible with neurobiological findings - assessing the validity of these arguments is beyond this reviewer's competence).

- elegance: the core idea is to push certain biologically implausible and computationally expensive parts of BPTT, which concern the forward propagation of effects of weight changes, into an *eligibility trace* of a synapse, a quantity that depends only on the past activity of this synapse and can be updated incrementally. This elegant insight is nicely summarized and highlighted at the beginning of the Discussion section (lines 322-331).

Relevance:

I (and the authors) consider the e-prop principle an important innovation in two respects:

- It provides a model mechanism that can explain how biological recurrent neural systems can achieve approximate gradient-based optimization. To my knowledge this is the first time that this is convincingly demonstrated. Efforts to find biological plausible (and computationally efficient) approximations to backpropagation training of neural networks are an important theme in neural network research, but have mostly been restricted to feedforward networks. Related other recent work in this direction is cited and fairly and adequately discussed in the Discussion section.

- e-prop may remove some painful roadblocks for the implementation of energy-efficient (due to spike-based operation) on-chip learning algorithms on novel spike-based neural network microchips, which constitute an increasingly important domain of microchip engineering. This potential impact is however discussed only in passing in the article, which is written more from a computational neuroscience perspective.

In summary I find this an extremely insightful and timely contribution to a research theme whose importance for computational neuroscience is obvious and whose importance for machine learning / microchip technology is becoming increasingly perceived.

A remaining question:

There is one aspect of this work which invites a critically minded discussion, and I would recommend that this point is more clearly addressed in the text, especially in the Discussion section.

This point that would deserve a more in-depth discussion concerns the replacement of the $dE/dh_j^{t'}$ factor in the classical BPTT formula (15) by heuristic „learning signals“ $L_j^{t'}$ which, unlike the original $dE/dh_j^{t'}$ factor, do not account for the forward propagation of the effects of weight changes. This is the reason why e-prop is only an approximation to BPTT. In the presented application demos this approximation does no apparent harm compared to BPTT. However I would think that there are other learning tasks where this approximation significantly degrades the performance. Learning tasks for (iterated) chaotic timeseries prediction might be particularly vulnerable to this approximation error due to the exponential blowup of perturbations in time in such systems. One might argue that such high-precision learning tasks are not what biological learning systems are made for, and neither are such tasks of great practical importance outside quite specific application domains like modeling fluid dynamics, for example. If one thinks further one might find other types of learning problems which critically depend on good approximations of the $dE/dh_j^{t'}$ factor. I would find it interesting if the authors would discuss what they think about the limitations incurred by the approximation of $dE/dh_j^{t'}$ by $L_j^{t'}$.

Two non-critical critical remarks:

1. From today's machine learning perspective it seems a little over-stretched to call the phoneme recognition demo a „speech recognition“ task. State of the art deep learning speech recognition solves far more involved learning problems and does not even use phoneme recognition as a subroutine (any longer). Specifically, today's deep learning speech recognition systems employ trainable attention mechanisms to integrate various kinds of information over multiple timescales in a mimicry of working memory mechanisms. This suggests (for future work) extensions of e-prop to multi-timescale (and likely multi-dimensional) eligibility traces, before broad-scale comparison claims with respect to BPPT can be made. The Discussion section would benefit from an appreciation of the role of multiple timescales.

2. I am always (not only in this manuscript) wary when a new method is benchmarked against

comparisons that are computed by the authors. A standard experience in machine learning (deep and otherwise) is that the performance of any basic method can be always improved if more care and time is applied to hyperparameter tuning, architecture refinement, regularization and data preprocessing which adapted to the used architecture and method. Thus I am normally only satisfied if comparisons are made against results published by the proponents of the competing methods, or if a documentation of the effort spent by the authors on optimizing the performance of the comparisons is given. Neither is the case for the phoneme recognition demo. But - oh, well - I'd let this pass.

Minor points:

Caption for Fig 1: last word: typography corrupt, at least in my pdf

Paragraph starting line 449: how is the pseudo-derivative defined for LIF neurons? the given formula is for ALIF neurons only. This information is also missing later in line 531 ff.

Line 473: denominator in fraction that starts this line should be ∂h_j^t

Line 477: denominator after = should read ∂h^t_j

Question: in Eq. (12), why not ... + $(1-\alpha) x^t$ on rhs?

Reviewer #2 (Remarks to the Author):

Learning in recurrent networks is intrinsically hard: deciding how to change a synapse requires considering both the direct local effects of the change on the impinging neuron, but also indirect effects via the network dynamics on all other neurons in the network, at all future time points. It is a fundamental challenge to understand how local plasticity mechanisms can effectively approximate this fundamentally non-local computation.

The manuscript by Bellec et al demonstrates a few successful examples of learning in recurrent spiking neural networks. This is achieved by combining a novel factorization of the traditional back-propagation through time (BPPT) algorithm (which allows for a local approximation of the gradients that drive learning) with a change in representation (via time-varying neural thresholds). While both of the contributions are potentially interesting, it remains conceptually unclear which of the two are the main factor that allow for the success demonstrated in the numerical simulations. The paper could greatly benefit from a more thorough explanation of the principles that make nontrivial learning possible in this model: less anecdotal evidence of success, more real intuition/ insight into biological learning.

Major concerns:

1. The factorization proposed here is an unusual hybrid between past facing and forward facing approaches to credit assignment. In the general form, I believe that the expression is not strictly correct, in that it double-counts each of the influences, once in the forward, and once in the backward pass. This is not an issue in the myopic approximation, but it does undermine the justification of the entire approach somewhat. This important point is largely obscured due to the notation by dropping the explicit time indexing of the loss terms.

2. As far as I can tell, the myopic approximation applied to vanilla spiking networks is strictly equivalent to Murray, 2019 (RFLO, ref39), something that the authors fail to acknowledge; this means that most improvements in performance come from the architecture, rather than the learning rule per se.

3. Since it is formally clear that the learning rule implemented here is a relatively coarse approximation to BPTT, it is puzzling why, when applied to their example tasks, the model performance ends up so close to exact BPTT. My suspicion is that although seemingly complex, the chosen tasks are simple in terms of spatio-temporal credit assignment demands, i.e. they don't really need the full BPTT power for learning to work. It would be useful to know more about the nature of the learned representations: how much the learned dynamics end up relying on recurrent connections for implementing the memory component of the task? Or is it that slow time constants due to the time-varying thresholds play the memory in this model? Is the network effectively feedforward such that there are no indirect effects of neural activity on the loss through recurrence? More generally, it would be useful to understand better how learning works. It may be helpful to take a very simple task with parametric variations in the constraints on credit assignment and work out the limits of the approximation in this space. Since it is an approximation, it necessarily breaks down at some point; it would be very helpful to know where that point is and what task parameters it depends on.

Minor concerns:

1. The framing of the current work in the context of the recent literature is fairly misleading:
A) in the intro, I found it strange having BP as the machine learning straw-man rather than BPPT, and postponing the discussion on the many other local approximations to BPTT to the discussion made it sound like is much more novel than warranted.
B) in the discussion, the comparison to other local temporal credit assignment approximations, in particular the stochastic vs deterministic and the BPTT vs RTRL distinctions seem directly lifted without credit from Marschall et al 2019. The authors cite the original papers directly — however the fact that they are approximations to RFLO was in no way acknowledged in the original papers and was only formally shown in the Marschall review, which is suspiciously missing from the references.
2. I found the ERN mentions confusing — the text lumps together underlying circuit mechanisms (DA) with indirect measurements of neural activity, which feels very unnatural from the perspective of experimental neuroscience.
3. “Without help from any teacher” (187) is odd given that the entire learning framework is a supervised one; granted the loss may be sparse over time, but that is a very different matter from lack of supervision.
4. The entire paragraph on neuromorphic chips 189-96 is out of place, maybe belongs in discussion, if at all.
5. It was unclear why the authors chose to add time varying thresholds in only a subset of neurons; how that affects the computation being performed and the nature of the solution?
6. Fig 3: there seems to be virtually no activity in the network during the delay: is the computation done exclusively in the thresholds? Potential links to Tsodyks-style W.Mem representations?
7. When comparing LIF vs time varying thresholds are the total numbers of parameters matched or just the number of neurons? Not sure which is the computationally more meaningful comparison... I would think the result somewhat trivial if the number of neurons is the same...

8. There is some confusion about time units, i.e. the learning framework is naturally defined in discrete time units, but it's unclear how does get mapped into biological time.

9. More details on the exact training procedures for the RL tasks would be helpful to understand conceptual differences relative to standard deep RL. There is potentially something very interesting in doing deep RL online but again, not enough intuition is provided to understand the contribution.

Typos and co:

1. It is strictly needed to add explicit time references in the equations, explicitly writing out the two time indexes corresponding to the application of the parameters and their effects at a later time
2. The use of partial derivatives unnecessarily complicates notation, while obscuring the main points; I would strongly advice replacing it with the more standard form

Reviewer #3 (Remarks to the Author):

This work addresses the problem of gradient-based training of recurrent spiking neural networks. As a solution, it proposes a biologically plausible local learning rule for spiking networks with threshold adaptation that involves an eligibility trace, pre- and postsynaptic activity. The novel learning rule allows solving complex tasks that involve temporal delays (e.g., atari games), which is even difficult for state-of-the-art learning schemes in the machine learning field and offers a biological (more) plausible way to implement (reinforcement) learning in spiking neural networks.

The presented results are of astounding importance both for theoretical neuroscience and for machine learning. Parts of the results have been presented in a similar form earlier (<https://openreview.net/forum?id=SkxJ4QKIIS>). However, here the results are explained in much greater detail, which significantly helps the understandability. Therefore I would enthusiastically recommend accepting the manuscript after some necessary revisions.

* To facilitate reproducibility, it is strongly desirable to make the code publicly available, e.g., on GitHub, preferentially already during the reviewing process.

* The spike trains after training with e-prop seem to be extremely regular clock-like (inside bursts of activity the coefficient of variation of the inter-spike interval seems very low.) The regular spiking can be seen e.g., in figure 3b, 4c, 5d, S1, S5a for random e-prop, Movie S4, Movie S5), Neurons in the cortex or hippocampus seem to have very irregular activity. Is the low irregularity (coefficient of variation of inter-spike intervals after training seems very small) a feature of the learning algorithm? If yes, how can this get more realistic irregular?

* Besides the display of spike trains, showing an example voltage trajectory and a histogram of voltages would corroborate the claim of biological realism.

* Are spikes in this framework necessary for computation, or are they just a biologically plausible feature that does not harm too much? If spikes are not required, could this be mapped to a rate-based analogous network with BCM-type plasticity, where analytical results might be easier to achieve?

* What are the core mechanisms of this learning algorithm, and how could they be understood in more detail?

* It is not clear how different simplifying assumptions in the approximation of the gradients are justified and why they have only a minor effect on the final performance. Are there certain limits where they break? If yes, when?

- * Moreover, the robustness of the results with respect to details of the parameters is not apparent.
- Is the excellent performance only observed in a small parameter regime that requires fine-tuning, or is it a general feature? When does it break down, and why?
- Is the assumption of a fully connected network crucial, or would this also work on sparse networks? (Section S2.4 claims it the evidence-accumulation task of figure 3 can still be solved with sparse networks obeying Dale's law, which would be very impressive, but it would be even more convincing to back this claim by adding the respective decision error to figure 3c).
- Is the assumption of a giant integration time-step of 1ms necessary, or would this also work on with $dt=0.05ms$ (as used usually in LIF model e.g., in [1])?
- * The central claim that e-prop "approaches the performance of BPTT" seems only partially be supported by numerical evidence. Errors after training with e-prop seem to decay much slower, and also, the final values seem much larger (e.g., more than a factor of 3 times larger in S5b). While the performance is still very impressive, the claim that it "approaches the performance of BPTT" does not seem to be justified. A log-log-scale in figure S5c would be helpful to compare the learning rates.
- * Are unphysiologically short membrane time constants (1ms) in the convolutional neural network necessary for the reinforcement tasks (Pong and Fishing Derby)? Maybe putting the network into a fluctuation-driven regime instead of a mean driven regime would yield sufficiently fast neurons with more realistic membrane time constant (10-20ms)?

Minor points:

- * The claim that e-prop is optimal seems only to concern the exponent of the computational complexity, but not the prefactor. This should be made clear.
- * Relating this work to previous attempts to train spiking neurons would be helpful. (e.g., D. Thalmeier, M. Uhlmann, B. Kappen, and R.-M. Memmesheimer 2016, DePasquale, B., Churchland, M.M. & Abbott, L.F. 2016, R. Guetig 2016, Kim, Chow 2018, A. Ingrosso, L.F. Abbott 2019, Gilra, A. & Gerstner, W. 2017, Alemi, A., Machens, C. K., Deneve, S. & Slotine, J.-J. 2018)
- * The study uses only one neuron model (LIF with/without adaptation), but it would be particularly exciting to discuss the limitation, e.g., would it also work with other commonly used spiking neuron models (e.g., exponential integrate-and-fire neurons, quadratic integrate-and-fire neurons)

[1].Renart, A. et al. The Asynchronous State in Cortical Circuits. *Science* 327, 587–590 (2010).

[2] Gilra, A. & Gerstner, W. Predicting non-linear dynamics by stable local learning in a recurrent spiking neural network. *eLife* 6, e28295 (2017).

[3] Alemi, A., Machens, C. K., Deneve, S. & Slotine, J.-J. Learning Nonlinear Dynamics in Efficient, Balanced Spiking Networks Using Local Plasticity Rules. in *Thirty-Second AAAI Conference on Artificial Intelligence* (2018).

Reviewer #1:

Competence of this reviewer:

Background in mathematics (dynamical systems) and RNNs, working in theoretical machine learning, only peripherally familiar with computational neuroscience.

Basis of this review:

I read the Introduction, Results, Discussion and the Methods up to (including) „Mathematical basis for e-prop“ in detail and only skimmed the rest due to lack of time. The „Mathematical basis for e-prop“ subsection of „Methods“ contains the core conceptual and mathematical innovation; the remaining parts which I only read superficially can be considered „mechanical“ workouts of the core idea and I trust the authors that this was done correctly.

With regard to “mechanical workouts of the core idea” we realized that we had not made sufficiently explicit that the application of e-prop to online deep RL in Fig. 4 and 5 requires an additional innovation: Previous solutions to Atari games with deep RL used either memory replay or simultaneous learning of several agents in order to avoid instabilities that typically arise when a single agent learns online with nonlinear function approximation. We are introducing a new –and biologically plausible– strategy where the agent first learns basic skills from shorter episodes with a high learning rate, and later refines these during longer episodes with a lower learning rate. We found that this simple strategy alleviates the previously mentioned instabilities. We had mentioned this already in Results (now on l 330 – 339) , but this remark of the reviewer motivated us to add a pointer to this in the Introduction (l 96 - 103).

Summary of contribution:

This work proposes a novel, approximate mathematical formulation of gradient descent optimization of recurrent neural networks (RNNs), called „e-prop“, and demonstrates the functionality of the new approach in nontrivial case studies (phoneme recognition, a maze navigation task with long short-term memory demands, Atari game learning) where it essentially comes out on a par with existing RNN training methods based on backpropagation through time (BPTT) and RNN-enhanced reinforcement learning. The main advantages of the novel formulation are

- generality: the scheme is applicable (with adequate specializations of basic formulas in each case) to many types of RNNs, including rate-based / LSTM networks and spiking variants; and to diverse sorts of loss functions (demonstrated in the paper: regression, classification, reinforcement learning)

- computational efficiency and biological plausibility: the new method obviates the biologically impossible and computationally expensive demands of BPTT for storing and processing many time-slice copies of the RNN. Weight adaptation is based only on information locally available at the concerned synapse, plus reward/teacher feedback information which needs no global integration of information (and is argued by the authors to be compatible with neurobiological findings - assessing the validity of these arguments is beyond this reviewer’s competence).

- elegance: the core idea is to push certain biologically implausible and computationally expensive parts of BPTT, which concern the forward propagation of effects of weight changes, into an *eligibility trace* of a synapse, a quantity that depends only on the past activity of this synapse and can be updated incrementally. This elegant insight is nicely summarized and highlighted at the beginning of the Discussion section (lines 322-331).

Relevance:

I (and the authors) consider the e-prop principle an important innovation in two respects:

- It provides a model mechanism that can explain how biological recurrent neural systems can achieve approximate gradient-based optimization. To my knowledge this is the first time that this is convincingly demonstrated. Efforts to find biological plausible (and computationally efficient) approximations to backpropagation training of neural networks are an important theme in neural network research, but have mostly been restricted to feedforward networks. Related other recent work in this direction is cited and fairly and adequately discussed in the Discussion section.

- e-prop may remove some painful roadblocks for the implementation of energy-efficient (due to spike-based operation) on-chip learning algorithms on novel spike-based neural network microchips, which constitute an increasingly important domain of microchip engineering. This potential impact is however discussed only in passing in the article, which is written more from a computational neuroscience perspective.

In summary I find this an extremely insightful and timely contribution to a research theme whose importance for computational neuroscience is obvious and whose importance for machine learning / microchip technology is becoming increasingly perceived.

A remaining question:

There is one aspect of this work which invites a critically minded discussion, and I would recommend that this point is more clearly addressed in the text, especially in the Discussion section.

This point that would deserve a more in-depth discussion concerns the replacement of the dE/dh_j^t factor in the classical BPTT formula (15) by heuristic „learning signals“ L_j^t which, unlike the original dE/dh_j^t factor, do not account for the forward propagation of the effects of weight changes. This is the reason why e-prop is only an approximation to BPTT. In the presented application demos this approximation does no apparent harm compared to BPTT. However I would think that there are other learning tasks where this approximation significantly degrades the performance. Learning tasks for (iterated) chaotic timeseries prediction might be particularly vulnerable to this approximation error due to the exponential blowup of perturbations in time in such systems. One might argue that such high-precision learning tasks are not what biological learning systems are made for, and neither are such tasks of great practical importance outside quite specific application domains like modeling fluid dynamics, for example. If one thinks further one might find other types of learning problems which critically depend on good approximations of the dE/dh_j^t factor. I would find it interesting if the authors would discuss what they think about the limitations incurred by the approximation of dE/dh_j^t by L_j^t .

This is a very good point, and we have added on l. 243 - 257 and 263 - 274 an analysis of this issue. We have also shown in Fig. S8 how one can break –with an admittedly somewhat artificial modification of the network architecture of Fig. 3– the capability of symmetric e-prop to approximate the performance level of BPTT.

We have also demonstrated in the new Fig. S8 the limitation of symmetric e-prop, as suggested by the reviewer.

Two non-critical critical remarks:

1. From today's machine learning perspective it seems a little over-stretched to call the phoneme recognition demo a „speech recognition“ task.

We agree, and refer now in the paper to this task as „phoneme recognition task“.

State of the art deep learning speech recognition solves far more involved learning problems and does not even use phoneme recognition as a subroutine (any longer). Specifically, today's deep learning speech recognition systems employ trainable attention mechanisms to integrate various kinds of information over multiple timescales in a mimicry of working memory mechanisms. This suggests (for future work) extensions of e-prop to multi-timescale (and likely multi-dimensional) eligibility traces, before broad-scale comparison claims with respect to BPPT can be made. The Discussion section would benefit from an appreciation of the role of multiple timescales.

This is a really fruitful suggestion for future work. We have added a corresponding remark on l 460 in the Discussion.

2. I am always (not only in this manuscript) wary when a new method is benchmarked against comparisons that are computed by the authors. A standard experience in machine learning (deep and otherwise) is that the performance of any basic method can be always improved if more care and time is applied to hyperparameter tuning, architecture refinement, regularization and data preprocessing which adapted to the used architecture and method. Thus I am normally only satisfied if comparisons are made against results published by the proponents of the competing methods, or if a documentation of the effort spent by the authors on optimizing the performance of the comparisons is given. Neither is the case for the phoneme recognition demo. But - oh, well - I'd let this pass.

For the TIMIT simulations we first reproduced the results from Greff et al. 2017 [ref. 23] and Graves et al. 2013 [ref. 24] with the exact same LSTM architectures as baselines. Only then we replaced the LSTM networks by LSNNs and BPTT by e-prop. We report the performance of these LSTM baselines in supplementary Figure S4. We now added the performance that was reported in those previous publications as dashed line.

For the ATARI benchmarks we had already reported the performance of the A3C paper (Mnih et al. 2016 [ref. 14]) in Figures 4 and 5. This appears to be the most comparable result from the literature. We have added to section S5.3 a new paragraph on “Details of evaluation” that compares the evaluation procedure of (Mnih et al, 2016) with ours. A pointer to that has been added in the last line of the caption to Fig. 4 in the main text.

Minor points:

Caption for Fig 1: last word: typography corrupt, at least in my pdf

Thanks. This was an artefact of the page layout, where the page number overlapped with the caption. We have fixed that.

Paragraph starting line 449: how is the pseudo-derivative defined for LIF neurons? the given formula is for ALIF neurons only. This information is also missing later in line 531 ff.

This information has been added on l 533 – 535. For LIF neurons the adaptive threshold A^t is replaced with its baseline value v_{th} .

Line 473: denominator in fraction that starts this line should be ∂h_j^t

Thanks, it has been corrected.

Line 477: denominator after = should read ∂h_j^t

Thanks, it has been corrected.

Question: in Eq. (12), why not $\dots + (1-\alpha) x^t$ on rhs?

We explain this on l 495 - 496. This simplifies the notation, and has no impact on performance if other parameters are scaled accordingly.

Reviewer #2 (Remarks to the Author):

Learning in recurrent networks is intrinsically hard: deciding how to change a synapse requires considering both the direct local effects of the change on the impinging neuron, but also indirect effects via the network dynamics on all other neurons in the network, at all future time points. It is a fundamental challenge to understand how local plasticity mechanisms can effectively approximate this fundamentally non-local computation.

The manuscript by Bellec et al demonstrates a few successful examples of learning in recurrent spiking neural networks. This is achieved by combining a novel factorization of the traditional back-propagation through time (BPPT) algorithm (which allows for a local approximation of the gradients that drive learning) with a change in representation (via time-varying neural thresholds). While both of the contributions are potentially interesting, it remains conceptually unclear which of the two are the main factor that allow for the success demonstrated in the numerical simulations. The paper could greatly benefit from a more thorough explanation of the principles that make nontrivial learning possible in this model: less anecdotal evidence of success, more real intuition/ insight into biological learning.

The applications that we had chosen appear to be natural next steps when one wants to train RSNNs to learn more difficult problems than previously possible with a biologically plausible learning method. In particular, we have chosen standard benchmark tasks on which the power of BPTT applied to LSTM networks had previously been demonstrated. The phoneme recognition task TIMIT is apparently one of the most frequently considered benchmark task for that, or more generally for temporal processing capability. For example, it had previously been used by one of the inventors of LSTM networks –Jürgen Schmidhuber– to probe in [Greff et al. 2017 = reference 23] the performance of BPTT for several versions of LSTM networks. We went in addition beyond this simpler framewise version of the TIMIT task from [Greff et al., 2017 =reference 23], and tested e-prop on the substantially more difficult sequence-based version of the task, that had been used as benchmark in the landmark paper [Graves et al., 2013 = reference 24]. Similarly, we had chosen Atari games as tasks for reward-based e-prop since these were used in the landmark paper [Mnih et al., 2016 = reference 14] as tasks for demonstrating the power of deep RL with BPTT. In addition, we have shown in the Supplement (Fig. S1 and S7) that the tasks that had been considered in the landmark paper [Nicola et al., Nature Communication 2017 = reference 35] on training RSNNs –with a method that was not argued to be biologically plausible– can also be solved by e-prop. We are not sure whether this relation of the benchmark tasks in our paper to the preceding literature had already been taken into account by the reviewer. We find that in view of this the wholesale negative evaluation of our experimental validation of e-prop (“demonstrates a few successful examples”, “anecdotal evidence of success”) is not justified.

We agree that additional explanations and intuitive understanding of e-prop and of the chosen architectures are helpful for the reader. Therefore, we have added on l. 243 - 257 and 263 - 274 an intuitive explanation of the approximation to BPTT that is carried out by e-prop. We have also introduced the non-technical but rather intuitive language “highways into the future” , see l 263 and the caption of movie S3, in order to provide an understanding of the role of eligibility traces in e-prop on a more intuitive level. These are contrasted with the “highways back into the past” –that BPTT uses by propagating signals backwards in time– in the caption of movie S2.

We have also added on l 45-51 and l 504 – 511 further explanations to the chosen RSNN model, as well as about its relation to biological data and a closely related neuron model from the Allen Institute. We had already shown in [3] that the capability of RSNNs without neurons that incorporate also slower processes of biological neurons –such as spike frequency adaptation– are not able to solve more demanding tasks such as phoneme recognition (TIMIT dataset) , even when one trains them with BPTT. Hence one cannot expect that e-prop can achieve that. Therefore, both a suitable RSNN model AND a powerful training method are needed for solving tasks such as phoneme recognition (TIMIT dataset). We have tested in addition in Fig. S3 whether recurrent connections are needed to solve this task well (they are needed). On the other hand, e-prop is obviously not tied to a particular RNN architecture. For example, it also works for LSTM networks, as we are showing in Fig. S4.

We hope that these additional explanations and figures provide an understanding of the relative contributions of the learning algorithms and architectures that are discussed in the paper.

Major concerns:

1. The factorization proposed here is an unusual hybrid between past facing and forward facing approaches to credit assignment. In the general form, I believe that the expression is not strictly correct, in that it double-counts each of the influences, once in the forward, and once in the backward pass. This is not an issue in the myopic approximation, but it does undermine the justification of the entire approach somewhat. This important point is largely obscured due to the notation by dropping the explicit time indexing of the loss terms.

Indeed the e-prop factorization in equation 1 combines a forward and backward credit assignment as illustrated in Figure 6 c. A detailed mathematical proof of equation 1 is given in the section „Mathematical basis for e-prop“. Hence we do not see a justification for the opinion of the reviewer that “the expression is not strictly correct”, or more specifically that it “double-counts” influences.

In addition to giving a rigorous mathematical proof in full detail, we also tested the validity of equation (1) numerically. The code for that can be found in the script `numerical_check_eprop_factorization_vs_BPTT.py`. There, we verified that when the learning signal L_j^t is computed in an exact fashion as $L_j^t = dE/dz_j^t$ through automatic differentiation, the sum of products $L_j^t e_{ji}^t$ is numerically equal to the gradient dE/dW computed through BPTT.

Having a time index on the loss would certainly make sense if the loss would decompose as a sum of instantaneous losses $E = \sum_t E^t$. But we have chosen to consider the general case where E does not necessarily decompose in this way. Hence we do not see ground for judging that “This important point is largely obscured due to the notation by dropping the explicit time indexing of the loss terms.”: We are not aware of a more precise notation that could possibly be used.

2. As far as I can tell, the myopic approximation applied to vanilla spiking networks is strictly equivalent to Murray, 2019 (RFLO, ref39), something that the authors fail to acknowledge; this means that most improvements in performance come from the architecture, rather than the learning rule per se.

The claim that one of our approaches “applied to vanilla spiking networks” would be “strictly equivalent” to a result of [Murray, 2019 = reference 51] is not correct: Murray’s paper provides only a learning method for non-spiking NNs. [Redacted] We also would like to point out that we had already given in the first version of our paper (now on l 430 – 433) a discussion of the relation of our work to his. In addition, we had checked beforehand whether James Murray would find this account adequate (he did).

Finally, we would like to mention that our first preprint on e-prop appeared on arxiv in January 2019, before the eLife paper by James Murray.

Altogether these facts suggest that there is no ground for the judgement of the reviewer that we “failed to acknowledge” the work of James Murray or others in our paper.

3. Since it is formally clear that the learning rule implemented here is a relatively coarse approximation to BPTT, it is puzzling why, when applied to their example tasks, the model performance ends up so close to exact BPTT. My suspicion is that although seemingly complex, the chosen tasks are simple in terms of spatio-temporal credit assignment demands, i.e. they don’t really

need the full BPTT power for learning to work. It would be useful to know more about the nature of the learned representations: how much the learned dynamics end up relying on recurrent connections for implementing the memory component of the task? Or is it that slow time constants due to the time-varying thresholds play the memory in this model? Is the network effectively feedforward such that there are no indirect effects of neural activity on the loss through recurrence? More generally, it would be useful to understand better how learning works. It may be helpful to take a very simple task with parametric variations in the constraints on credit assignment and work out the limits of the approximation in this space. Since it is an approximation, it necessarily breaks down at some point; it would be very helpful to know where that point is and what task parameters it depends on.

We would like to thank the reviewer for these suggestions. We have added an analysis of the functional role of recurrent connections for the first task –TIMIT– in the new Fig. S3. The result shows that recurrent connections are essential for solving this task well with a spiking network.

On the other hand, we have shown in the new Fig. S8 that adding a feedforward network to the architecture of the recurrent network for solving the task of Fig. 3 breaks the good performance of symmetric and adaptive e-prop, and reduces the reliability of random e-prop. These results suggest that e-prop is a learning method that is specially suited for training recurrent SNNs, rather than feedforward SNNs. We believe that both recurrent connections and slow time constants are essential for the capability of e-prop to approximate the performance of BPTT, see also our new explanations on l 243 – 283.

Minor concerns:

1. The framing of the current work in the context of the recent literature is fairly misleading: A) in the intro, I found it strange having BP as the machine learning straw-man rather than BPPT, and postponing the discussion on the many other local approximations to BPTT to the discussion made it sound like it is much more novel than warranted.

„BPTT“ was mentioned 7 times in the Introduction, whereas BP is only mentioned once –in order to explain how BPTT works. Hence the criticism of the reviewer that we are „having BP as the machine learning straw-man rather than BPPT“ is incorrect. The reviewer uses this incorrect evidence as the sole evidence for the sweeping claim that “The framing of the current work in the context of the recent literature is fairly misleading”.

Our submission contained on lines 348 – 387 a thorough discussion of the relation of our work to previous related work. We found it would overload the Introduction if this extensive discussion would be moved to the Introduction. However, we have added now on l 104 of the Introduction a pointer to this discussion:

„We are not aware of previous work on online gradient descent learning methods for RSNNs, neither for supervised learning nor for RL. There exists however preceding work about online approximations of gradient descent for non-spiking neural networks, which is reviewed in the Discussion section.” Finally, we would like to appeal to Reviewer 2 to recognize that the topic of this paper is learning in recurrent networks of SPIKING neurons, see its title abstract, and content.

We also would like to mention that –in contrast to Reviewer 2– Reviewer 1 found that „Related other recent work in this direction is cited and fairly and adequately discussed in the Discussion section“.

B) in the discussion, the comparison to other local temporal credit assignment approximations, in particular the stochastic vs deterministic and the BPTT vs RTRL distinctions seem directly lifted without credit from Marschall et al 2019. The authors cite the original papers directly — however the fact that they are approximations to RFLO was in no way acknowledged in the original papers and was only formally shown in the Marschall review, which is suspiciously missing from the references.

We were not aware of this review by Marschall, Cho, and Savin. Apparently the reviewer wants that we cite this review, and we have included therefore on l 425 a reference. But we propose that it makes sense to also cite the original papers directly. The distinction between stochastic and deterministic approximation methods and BPTT vs RTRL in our discussion of preceding work was dictated by the methods that had previously been used. We also cannot understand why anyone would want to steal such a distinction from a review paper without citing it, as the reviewer insinuates.

2. I found the ERN mentions confusing — the text lumps together underlying circuit mechanisms (DA) with indirect measurements of neural activity, which feels very unnatural from the perspective of experimental neuroscience.

We agree that ERN is an indirect measurement of neural activity. Therefore, we have reformulated the corresponding sentences in the introduction (now on l 74) in order to make our reference to the paper (Sajad et al., 2019) [11] from experimental neuroscience more explicit, which reports measurements of neural activity that is directly related to the ERN.

3. “Without help from any teacher” (l87) is odd given that the entire learning framework is a supervised one; granted the loss may be sparse over time, but that is a very different matter from lack of supervision.

We do not agree with the reviewer that “the entire learning framework is a supervised one”. Rather, we introduce methods for reducing a given loss function by gradient descent through biologically plausible mechanisms. This loss function can be based on teacher input in the case of supervised learning. But for applications to RL there is no teacher input or “supervision” involved in the loss function. One can in fact also apply e-prop in the absence of both a teacher and reward, to purely unsupervised learning, as we will show in a follow-up paper. This is just a matter of defining the loss function in a suitable manner

A characterization of reinforcement learning (RL) as learning „without any help from a teacher“ is in our view fairly standard RL, see e.g. (Sutton and Barto, 2018). We have added on l 89-96 a more detailed explanation for readers who are less familiar with the RL literature.

4. The entire paragraph on neuromorphic chips 189-96 is out of place, maybe belongs in discussion, if at all.

An Introduction should sketch the scientific background and context of a paper. Since the exploration of online learning methods for RSNNs is a very important and active research area also in neuromorphic engineering, and since our results directly contribute to this area – e-prop is currently implemented on the neuromorphic chip SpiNNaker and the next generation of Intel’s neuromorphic chip Loihi– we found that this link to neuromorphic engineering should be mentioned in the Introduction.

We also would like to point out that Reviewers 1 and 3 emphasized the relevance of this link. Therefore, we propose to keep these lines in the Introduction as is.

5. It was unclear why the authors chose to add time varying thresholds in only a subset of neurons; how that affects the computation being performed and the nature of the solution?

Good point. We added an explanation from the perspective of biological data on p. 504 – 506 and p. 523 – 525.

We agree that a thorough analysis of the computational impact of the size of the fraction of neurons that exhibit spike frequency adaptation is an interesting topic. In preliminary analyses we found that the precise value of this fraction has little impact on computational performance. Since this topic appears to be somewhat marginal for this paper, we will address this issue in a separate paper.

6. Fig 3: there seems to be virtually no activity in the network during the delay: is the computation done exclusively in the thresholds? Potential links to Tsodyks-style W.Mem representations?

We also believe that the time-varying thresholds carry important information, even in the absence of firing activity. One remark on that has been added at the end of S3.4 in the Supplement: “Analysis of information in adaptive thresholds”. We agree that RSNNs with adaptive thresholds can be seen as providing models for activity-silent working memory, somewhat analogous to the model of Tsodyks et al. but based on different mechanism and different biological data. We plan to address this issue in a separate paper.

7. When comparing LIF vs time varying thresholds are the total numbers of parameters matched or just the number of neurons? Not sure which is the computationally more meaningful comparison... I would think the result somewhat trivial if the number of neurons is the same...

Perhaps the reviewer refers here to the analysis of the number of parameters that are tuned in different learning models. This number is the same for LIF neurons and for neurons with time-varying thresholds, since in either case only the synaptic weights are tuned during learning.

8. There is some confusion about time units, i.e. the learning framework is naturally defined in discrete time units, but it's unclear how does get mapped into biological time.

We had to carry out our simulations on digital computers, hence in discrete time. But we have now carried out additional control simulations for the task of Fig. 3 with smaller discrete time steps than our standard 1 ms –up to the limit imposed by our computing resources. The results are reported in Fig. S6. They suggest that the size of the discrete time step has no salient impact on the performance of e-prop for this task. We have added a corresponding remark on l 485 - 487.

9. More details on the exact training procedures for the RL tasks would be helpful to understand conceptual differences relative to standard deep RL. There is potentially something very interesting in doing deep RL online but again, not enough intuition is provided to understand the contribution.

We have described a salient algorithmic difference of our RL approach to standard deep RL on l 96 – 103. Furthermore we have described on l 292 – 296 a particular feature of the e-prop implementation of deep RL. We have also added in the subsection “Details of the learning procedure” in section S5.3 further details of our learning procedure for deep RL, and its difference to the previous standard.

Typos and co:

1. It is strictly needed to add explicit time references in the equations, explicitly writing out the two time indexes corresponding to the application of the parameters and their effects at a later time

We tried to make the time index explicit throughout the paper. We would be grateful for a pointer if one is missing.

2. The use of partial derivatives unnecessarily complicates notation, while obscuring the main points; I would strongly advice replacing it with the more standard form

We believe that our notation is standard, and that there are no viable alternatives on the basis of standard literature. Our notation is for example consistent with the wikipedia article on total derivatives (see examples of section 3. Chain rule with total derivatives https://en.wikipedia.org/wiki/Total_derivative).

Some authors do not distinguish partial and total derivatives and interchange the “rounded d” and the “sharp d”. In our analysis it was crucial to make this distinction clear and we believe that we took the most standard notation for that. We have described the standard convention that we are using on l 549 – 558.

Reviewer #3:

This work addresses the problem of gradient-based training of recurrent spiking neural networks. As a solution, it proposes a biologically plausible local learning rule for spiking networks with threshold adaptation that involves an eligibility trace, pre- and postsynaptic activity. The novel learning rule allows solving complex tasks that involve temporal delays (e.g., atari games), which is even difficult for state-of-the-art learning schemes in the machine learning field and offers a biological (more) plausible way to implement (reinforcement) learning in spiking neural networks.

The presented results are of astounding importance both for theoretical neuroscience and for machine learning. Parts of the results have been presented in a similar form earlier (<https://openreview.net/forum?id=SkxJ4QKIIS>). However, here the results are explained in much greater detail, which significantly helps the understandability. Therefore I would enthusiastically recommend accepting the manuscript after some necessary revisions.

* To facilitate reproducibility, it is strongly desirable to make the code publicly available, e.g., on GitHub, preferentially already during the reviewing process.

The code is made available to the reviewers as zip file, and we will be publish it on github https://github.com/IGITUGraz/eligibility_propagation as soon a the paper is accepted. It includes the code that reproduces all the experiments discussed in the main text as shown in Figures 2,3,4 and 5. This repository is announced to the reader on l 741 – 747 .

* The spike trains after training with e-prop seem to be extremely regular clock-like (inside bursts of activity the coefficient of variation of the inter-spike interval seems very low.) The regular spiking can be seen e.g., in figure 3b, 4c, 5d, S1, S5a for random e-prop, Movie S4, Movie S5), Neurons in the cortex or hippocampus seem to have very irregular activity. Is the low irregularity (coefficient of variation of inter-spike intervals after training seems very small) a feature of the learning algorithm? If yes, how can this get more realistic irregular?

Good point. We are showing in the revised versions of Fig. 3, 4, 5 and in the new Fig. S9 that regular clock-like spiking activity can be avoided by including a regularization term in the loss function that prefers low values of membrane voltages. This leads to sparser firing, and resulting traces of membrane voltages become more similar to traces from neural recordings. The technical details of this regularization are described in section S2.4 of the Supplement. Altogether, we propose that regularity of spike trains is not a consequence of the e-prop learning method, but of the regularization terms that are used in the loss function.

* Besides the display of spike trains, showing an example voltage trajectory and a histogram of voltages would corroborate the claim of biological realism.

We show example voltage traces in the new version of Fig. 3b, Fig. 4c, and further voltage traces together with a histogram in Fig. S9.

* Are spikes in this framework necessary for computation, or are they just a biologically plausible feature that does not harm too much? If spikes are not required, could this be mapped to a rate-based analogous network with BCM-type plasticity, where analytical results might be easier to achieve?

We are not able to provide evidence that spikes are necessary for computation. But our results do show on the side that powerful computing and learning performance is possible with sparse firing in SNNs, hence potentially at a much lower energy consumption than with clocked ANNs –which may be important both for brains and adaptive edge devices in technology.

We show in Fig. S4 that e-prop can in principle also be applied to non-spiking NNs with slowly changing hidden variable such as LSTM networks. We had reported further applications of e-prop to non-spiking NNs in Fig. 4 of our first report on e-prop from January 2019 in arxiv. But we felt that these results would be beyond the scope of this paper. We agree that it would be very interesting to compare e-prop with BCM-type plasticity, but felt that this should rather be left to future work.

* What are the core mechanisms of this learning algorithm, and how could they be understood in more detail?

Good point! We have added on l 243 - 283 an account of the core mechanism behind e-prop, including when it is likely to work well, and if so, why. We have added in Fig. S8 results of corresponding control experiments (with technical details provided in the subsection “Details of the control experiment in Fig. S8” in section S3.4 of the Supplement).

* It is not clear how different simplifying assumptions in the approximation of the gradients are justified and why they have only a minor effect on the final performance. Are there certain limits where they break? If yes, when?

We have tried to answer this with the same additional text and control experiment as specified for the preceding point. In particular, our text on l266 – 283 and Fig. S8 summarize our knowledge about when specific approximations of gradients break.

* Moreover, the robustness of the results with respect to details of the parameters is not apparent. - Is the excellent performance only observed in a small parameter regime that requires fine-tuning, or is it a general feature? When does it break down, and why?

We report for the experiment of Fig. 3 in panel c average performance over 20 runs with different initializations, and also results for different versions of e-prop, and also for e-prop in combination with adaptive rewiring. Similarly, for reward-based RL we report in Fig. 4 and 5 average performance over different runs with different initialization. In Fig. S3 we report for TIMIT control results for different network architectures. In general, we had not fine-tuned hyperparameters to achieve good performance. This gave us the impression that learning via e-prop is in general less parameter sensitive than many other approaches in computational neuroscience. Since we are now making the

code accessible to the reviewers (and upon publication of the paper to the public) this issue can be further explored by the scientific community.

- Is the assumption of a fully connected network crucial, or would this also work on sparse networks? (Section S2.4 claims it the evidence-accumulation task of figure 3 can still be solved with sparse networks obeying Dale's law, which would be very impressive, but it would be even more convincing to back this claim by adding the respective decision error to figure 3c).

We found that the assumption of a fully connected network is not crucial, and added –as suggested by the reviewer– to Fig. 3c a performance curve for a sparsely connected network that obeys Dale's law. Even though the network learns slower, it can still solve the task. This new result is commented on l 238 – 240, with details in section S2.5 of the Supplement.

- Is the assumption of a giant integration time-step of 1ms necessary, or would this also work on with $dt=0.05ms$ (as used usually in LIF model e.g., in [1])?

We tested this by running the temporal credit assignment task of Fig. 3 with different step sizes down to $dt=0.05$, as requested by the reviewer. The results are shown in Fig. S6, and explained in the caption of that figure. We also added a pointer to this in l 485 – 487 of the main text. One sees that learning performance has no visible dependence on the simulation step size, but that the wall clock time of the simulation increases drastically with smaller step sizes.

* The central claim that e-prop "approaches the performance of BPTT" seems only partially be supported by numerical evidence. Errors after training with e-prop seem to decay much slower, and also, the final values seem much larger (e.g., more than a factor of 3 times larger in S5b). While the performance is still very impressive, the claim that it "approaches the performance of BPTT" does not seem to be justified. A log-log-scale in figure S5c would be helpful to compare the learning rates.

We agree that e-prop learns slower than BPTT, and added a remark in l 82-83 to make this clear. We also added the suggested plot in log-log scale to figure S7 (formerly figure S5). Our goal was to convey the fact that the resulting network performance after e-prop learning is usually within a few percent of the performance achieved with training by BPTT, as demonstrated in Fig. 2c, Fig. 3c, Fig. 4d, Fig. 5c, S1 b, S2, S3, S4, S5. We thought that the term "approaches the performance of BPTT" would be a suitable way of describing that. In particular, compared with previous approaches towards biologically plausible learning in RSNNs it appears to provide a qualitative jump with regard to getting closer to the performance of BPTT. We also believe that the performance of e-prop can be further improved with better online learning signals, as for example the ones that are produced by a separately trained RSNN in the Learning-to-Learn approach of Fig. 3 of our first report on e-prop from January 2019 in arxiv. We will address this in a follow-up paper.

* Are unphysiologically short membrane time constants (1ms) in the convolutional neural network necessary for the reinforcement tasks (Pong and Fishing Derby)? Maybe putting the network into a fluctuation-driven regime instead of a mean driven regime would yield sufficiently fast neurons with more realistic membrane time constant (10-20ms)?

We have explained the reason for this unphysiologically short time constant in the subsection “Details of the network models” of section S5.2. In short, the limitation of the compute time that was available to us forced us to work with an extremely high frame rate of screen shots from the Atari game as network inputs. This required then a very short membrane time constant in the CNN for achieving good performance.

Please note that the spiking CNN was not supposed to model biological vision in an accurate way. Rather, it was just a way to make the high-dimensional input accessible to the RSNN.

Minor points:

* The claim that e-prop is optimal seems only to concern the exponent of the computational complexity, but not the prefactor. This should be made clear.

We agree, and have made this explicit on l 435 in the Discussion.

* Relating this work to previous attempts to train spiking neurons would be helpful. (e.g., D. Thalmeier, M. Uhlmann, B. Kappen, and R.-M. Memmesheimer 2016, DePasquale, B., Churchland, M.M. & Abbott, L.F. 2016, R. Guetig 2016, Kim, Chow 2018, A. Ingrosso, L.F. Abbott 2019, Gilra, A. & Gerstner, W. 2017, Alemi, A., Machens, C. K., Deneve, S. & Slotine, J.-J. 2018)

We have added these references (with the exception of Guetig et 2016, since it discussed only plasticity in a single neuron, and we would have to add many further references for a proper review of this topic). We have discussed their contributions and the relationship to eprop on l 401 – 408.

* The study uses only one neuron model (LIF with/without adaptation), but it would be particularly exciting to discuss the limitation, e.g., would it also work with other commonly used spiking neuron models (e.g., exponential integrate-and-fire neurons, quadratic integrate-and-fire neurons)

Our approach can obviously be applied to any differentiable neuron model, since the theoretical support that we have presented does not depend on the neuron model. The case of a spiking neuron shows that it is in fact also applicable to non-differentiable neuron models, but one needs to identify suitable pseudo-derivatives for that (we are for example currently working on an extension to stochastic spiking neuron models). We have added a remark on this on l 441 - 442 of the Discussion.

REVIEWERS' COMMENTS:

Reviewer #1 (Remarks to the Author):

Reviewer 1: Review of revision 1 of „A solution to the learning dilemma for recurrent networks of spiking neurons“ from Bellec et al.

Summary assessment: I find that the authors spent the right amount of work and even some more than that (new figures, extended experimentation) to address the reviewer's comments from round 1.

When re-reading the paper (I did not review the supplements, having essentially no free time) I found that the reading experience could be made much smoother by adding short explanations at a few decisive points in the text. I pointed this out by annotating the manuscript pdf - my comments may be best seen using Acrobat Reader (not all pdf readers show the annotations that I made). I attach the annotated manuscript (my identity is anonymized).

Finally, the style of English is „germanic“ in some places. If journal editors take care of that, super. If nothing is done, also good enough. If the authors want to englishify the English themselves - I marked some sentences in the annotated pdf which would benefit from rephrasing.

Reviewer #2 (Remarks to the Author):

I find myself conflicted about this manuscript. It is not a bad piece of work, but it could be much better.

I still believe there is a fundamental conceptual confusion in the presentation of the results in that the text mashes together 2 separate issues:

- 1) the choice of architecture (where here I meant not LSTMs/GRUs but the use of spiking neurons with time varying thresholds, since such neuron models have been previously formalized as 2 layer ANNs, see e.g. Zenke review)
and
- 2) the issue of approximating BPPT gradients in a new way, which is what makes the explicit time indexing of the gradient terms and the comparison to other BPPT approximations critical for understanding the update rule and for putting the work in context.

Now there is a 3rd point of emphasis on the use of the rule for RL and the shaping procedure required to train networks for such tasks. This is also potentially interesting but not explored enough to yield true insight, leaving the point somewhat anecdotal.

The text changed very little to address any of the concerns I had raised.

The focus remains on numerical results rather than a true understanding of the underlying processes, which is generally OK for an application centric machine learning paper but less so for computational neuroscience research, aimed at formulating new theories about how the brain works.

Reviewer #3 (Remarks to the Author):

The authors have done an excellent job in addressing most of my concerns raised during the first review. I have one major concern and a list of minor issues:

Remaining major concern:

* For the adaptive LIF, the adaptation time constant is "chosen to be in the range of the time span of the length of the working memory that is a relevant for a given task". Based on the Technical White paper on the generalized LIF (GLIF) model, it is not clear if neurons in the brain have such a slow threshold adaptation (on the order of seconds). A reference to e.g. fitted data from the Allen brain atlas that is consistent with adaptation constants on the order of seconds would be helpful to support the claim of a biologically realistic neuron model. I wasn't able to find supporting evidence, but I am also not an expert on this specific question.

* The code was made available, and the reviewer was able to reproduce figure 3. (Minor point: The script generated the previous version of the figure without voltage trace).

* The issue of unrealistic regular clock-like spiking was addressed: by including a regularization term in the loss which imposes a soft constraint on the membrane potentials, the authors found an ad-hoc solution to generate more irregular and sparse firing patterns (figure 3b, 4c). While this ad-hoc solution seems not to be biologically realistic, it quite convincingly shows that unrealistic regular spiking is not a necessary consequence of the novel introduced learning method e-prop. Thus, this issue is addressed adequately. While spikes seem not necessary for computation in this framework, the work demonstrates that it is possible to train spiking networks on difficult tasks despite the difficulties arising by spikes, which is very impressive.

* a heuristic explanation of the learning algorithm's core mechanisms is given in more detail in the main manuscript, which significantly helps in understanding e-prop more intuitively.

* Moreover, additional numerical experiments (S.8) are performed, which are consistent with the explanation in terms of gradient information available online (in random and symmetric e-prop the partial derivative $\partial E / \partial z$ are used in contrast to usual backpropagation through time where weights are adapted based on the gradient coming from the total derivative dE/dz). This also addresses when e-prop is expected to succeed and fail, which is a vital improvement of the manuscript.

* Thanks to the code's availability, the risk of fine-tuned (hyper) parameters is drastically reduced and spot check simulation of figure 3 with slightly varied parameters resulted in similar results. The reviewer noted however that the refractory period of 5ms seems larger and reducing it results for figure 3 in spike bursts whose spike rate is limited only by the inverse of the refractory period, which seems to indicate that the spiking network is not in a fluctuation-driven regime, but operates in a mean-driven regime.

* The addition in figure 3c that the training still works after constraining the network to obey Dale's law is remarkable.

* the additional analysis in different time-discretizations convinced the reviewer that e-prop also works with smaller integration time-step.

* The performance comparison (both in words and in numbers) is in the revised version fairer, more nuanced, and adequate to the actual presented numbers.

* The reviewer has to apologize that he missed the discussion of the short time constant in the subsection "Details of the network models" of section S5.2. It seems to be justified in this case.

* The reviewer agrees with the decisions considering references and the discussion of previous approaches to train RNNs.

Minor points:

- * line 408: the last ":" before e-prop is not necessary.
- * "best know" should be "best-known"
- * "a feedforward deep network" should be "a deep feedforward network"
- * "in deep learning this problem" should be "in deep learning, this problem"
- * "in machine learning one trains recurrent" should be "in machine learning, one trains .."
- * "in the brain there exists an abundance of top-down" should be "in the brain, there exists an abundance of top-down"
- * "Furthermore dopamine signals" should be "Furthermore, dopamine signals"
- * "In such reinforcement learning (RL) tasks the learner needs to explore its environment, and find.." should be "In such reinforcement learning (RL) tasks, the learner needs to explore its environment and find.."
- * "Nevertheless learning methods" should be "Nevertheless, learning methods" should be"
- * "There one trains" should be "There, one trains"
- * "neural networks of the brain, since" should be "neural networks of the brain since"
- * Always "However,"
- * "such as for example" should be "such as"
- * "Furthermore," ..
- * "RSNNs which are attainable" should be "RSNNs, which are attainable"
- * "In adaptive e-prop we let" should be "In adaptive e-prop, we let"
- * "In particular they involve" should be "In particular, they involve"
- * "neuron specific" should be "neuron-specific"
- * "Furthermore LSNNs" should be "Furthermore, LSNNs"
- * "energy efficient" should be "energy-efficient"
- * "provides also" should be "also provides"
- * "Furthermore the cues" should be "Furthermore, the cues"
- * "if all the learning signals are identically until the last" should be "if all the learning signals are identical until the last"
- * "In symmetric and adaptive e-prop one uses the partial" should be "In symmetric and adaptive e-prop, one uses the partial"
- * "difficult since" should be "difficult, since"
- * "Currently we are not" should be "Currently, we are not"
- * "abundantly" should be "abundantly"
- * "target specific" should be "target-specific"
- * "Instead some" should be "Instead, some"
- * "But in a recurrent network the" should be "But in a recurrent network, the"
- * "which in turn affect firing" should be "which in turn affect the firing"
- * "within the reach of" should be "within reach of"
- * "one dimensional internal state" should be "one-dimensional internal state"
- * "occured" should be "occurred"
- * "For simplicity we have" should be "For simplicity, we have"
- * "as large as the adaptation adaptation time constant" should be "as large as the adaptation time constant"
- * "can be found in the section" "can be found in section"
- * "Until now this derivation follows" should be "Until now, this derivation follows"

Reviewer #1:

Summary assessment: I find that the authors spent the right amount of work and even some more than that (new figures, extended experimentation) to address the reviewer's comments from round 1.

When re-reading the paper (I did not review the supplements, having essentially no free time) I found that the reading experience could be made much smoother by adding short explanations at a few decisive points in the text. I pointed this out by annotating the manuscript pdf - my comments may be best seen using Acrobat Reader (not all pdf readers show the annotations that I made). I attach the annotated manuscript (my identity is anonymized).

Finally, the style of English is „germanic“ in some places. If journal editors take care of that, super. If nothing is done, also good enough. If the authors want to englishify the English themselves - I marked some sentences in the annotated pdf which would benefit from rephrasing.

Thank you, we have rephrased these sentences.

Reviewer #2:

I find myself conflicted about this manuscript. It is not a bad piece of work, but it could be much better.

I still believe there is a fundamental conceptual confusion in the presentation of the results in that the text mashes together 2 separate issues:

- 1) the choice of architecture (where here I meant not LSTMs/GRUs but the use of spiking neurons with time varying thresholds, since such neuron models have been previously formalized as 2 layer ANNs, see e.g. Zenke review)
and
- 2) the issue of approximating BPPT gradients in a new way, which is what makes the explicit time indexing of the gradient terms and the comparison to other BPPT approximations critical for understanding the update rule and for putting the work in context.

Now there is a 3rd point of emphasis on the use of the rule for RL and the shaping procedure required to train networks for such tasks. This is also potentially interesting but not explored enough to yield true insight, leaving the point somewhat anecdotal.

The text changed very little to address any of the concerns I had raised. The focus remains on numerical results rather than a true understanding

of the underlying processes, which is generally OK for an application centric machine learning paper but less so for computational neuroscience research, aimed at formulating new theories about how the brain works.

Reviewer #3:

The authors have done an excellent job in addressing most of my concerns raised during the first review. I have one major concern and a list of minor issues:

Remaining major concern:

* For the adaptive LIF, the adaptation time constant is "chosen to be in the range of the time span of the length of the working memory that is a relevant for a given task". Based on the Technical White paper on the generalized LIF (GLIF) model, it is not clear if neurons in the brain have such a slow threshold adaptation (on the order of seconds). A reference to e.g. fitted data from the Allen brain atlas that is consistent with adaptation constants on the order of seconds would be helpful to support the claim of a biologically realistic neuron model. I wasn't able to find supporting evidence, but I am also not an expert on this specific question.

The published data from the Allen Institute on spike frequency adaptation do not provide information about the duration of this effect, since they only carried out short experiments. We therefore added in the section on LSNNs in Methods a reference to

Pozzorini, C., Naud, R., Mensi, S., & Gerstner, W. (2013). Temporal whitening by power-law adaptation in neocortical neurons. *Nature neuroscience*, 16(7), 942.

There it is reported on p. 946 that the adaptation effect lasted for 20s in some neurons.

* The code was made available, and the reviewer was able to reproduce figure 3. (Minor point: The script generated the previous version of the figure without voltage trace).

* The issue of unrealistic regular clock-like spiking was addressed: by including a regularization term in the loss which imposes a soft constraint on the membrane potentials, the authors found an ad-hoc solution to generate more irregular and sparse firing patterns (figure 3b, 4c). While this ad-hoc solution seems not to be biologically realistic, it quite convincingly shows that unrealistic regular spiking is not a necessary consequence of the novel introduced learning method e-prop. Thus, this issue is addressed adequately. While spikes seem not necessary for computation in this framework, the work demonstrates that it is possible to train spiking networks on difficult tasks despite the difficulties arising by spikes, which is very impressive.

* a heuristic explanation of the learning algorithm's core mechanisms is given in more detail in the main manuscript, which significantly helps in understanding e-prop more intuitively.

* Moreover, additional numerical experiments (S.8) are performed, which are consistent with the explanation in terms of gradient information available online (in random and symmetric e-prop the partial derivative $\partial E / \partial z$ are used in contrast to usual backpropagation through time where weights are adapted based on the gradient coming from the total derivative dE/dz). This also addresses when e-prop is expected to succeed and fail, which is a vital improvement of the manuscript.

* Thanks to the code's availability, the risk of fine-tuned (hyper) parameters is drastically reduced and spot check simulation of figure 3 with slightly varied parameters resulted in similar results. The reviewer noted however that the refractory period of 5ms seems larger and reducing it results for figure 3 in spike bursts whose spike rate is limited only by the inverse of the refractory period, which seems to indicate that the spiking network is not in a fluctuation-driven regime, but operates in a mean-driven regime.

* The addition in figure 3c that the training still works after constraining the network to obey Dale's law is remarkable.

* the additional analysis in different time-discretizations convinced the reviewer that e-prop also works with smaller integration time-step.

* The performance comparison (both in words and in numbers) is in the revised version fairer, more nuanced, and adequate to the actual presented numbers.

* The reviewer has to apologize that he missed the discussion of the short time constant in the subsection "Details of the network models" of section S5.2. It seems to be justified in this case.

* The reviewer agrees with the decisions considering references and the discussion of previous approaches to train RNNs.

Minor points:

Thank you; these suggestions have all been implemented.

* line 408: the last ":" before e-prop is not necessary.

* "best know" should be "best-known"

* "a feedforward deep network" should be "a deep feedforward network"

* "in deep learning this problem" should be "in deep learning, this problem"

- * "in machine learning one trains recurrent" should be "in machine learning, one trains .."
- * "in the brain there exists an abundance of top-down" should be "in the brain, there exists an abundance of top-down"
- * "Furthermore dopamine signals" should be "Furthermore, dopamine signals"
- * "In such reinforcement learning (RL) tasks the learner needs to explore its environment, and find.." should be "In such reinforcement learning (RL) tasks, the learner needs to explore its environment and find.."
- * "Nevertheless learning methods" should be "Nevertheless, learning methods" should be"
- * "There one trains" should be "There, one trains"
- * "neural networks of the brain, since" should be "neural networks of the brain since"
- * Always "However,"
- * "such as for example" should be "such as"
- * "Furthermore," ..
- * "RSNNs which are attainable" should be "RSNNs, which are attainable"
- * "In adaptive e-prope we let" should be "In adaptive e-prop, we let"
- * "In particular they involve" should be "In particular, they involve"
- * "neuron specific" should be "neuron-specific"
- * "Furthermore LSNNs" should be "Furthermore, LSNNs"
- * "energy efficient" should be "energy-efficient"
- * "provides also" should be "also provides"
- * "Furthermore the cues" should be "Furthermore, the cues"
- * "if all the learning signals are identically until the last" should be "if all the learning signals are identical until the last"
- * "In symmetric and adaptive e-prop one uses the partial" should be "In symmetric and adaptive e-prop, one uses the partial"
- * "difficult since" should be "difficult, since"
- * "Currently we are not" should be "Currently, we are not"
- * "abundantly" should be "abundantly"
- * "target specific" should be "target-specific"
- * "Instead some" should be "Instead, some"
- * "But in a recurrent network the" should be "But in a recurrent network, the"
- * "which in turn affect firing" should be "which in turn affect the firing"
- * "within the reach of" should be "within reach of"
- * "one dimensional internal state" should be "one-dimensional internal state"
- * "occured" should be "occurred"
- * "For simplicity we have" should be "For simplicity, we have"
- * "as large as the adaptation adaptation time constant" should be "as large as the adaptation time constant"
- * "can be found in the section" "can be found in section"
- * "Until now this derivation follows" should be "Until now, this derivation follows"